# From Heuristic to Analytic: Cognitively Motivated Strategies for Coherent Physical Commonsense Reasoning

**Zheyuan Zhang**[1*]      **Shane Storks**[1*]      **Fengyuan Hu**[1]      **Sungryull Sohn**[2]
**Moontae Lee**[2,3]      **Honglak Lee**[1,2]      **Joyce Chai**[1]
[1]University of Michigan      [2]LG AI Research
[3]University of Illinois at Chicago
{zheyuan, sstorks, hufy, chaijy}@umich.edu
{srsohn, moontae.lee, honglak}@lgresearch.ai

## Abstract

Pre-trained language models (PLMs) have shown impressive performance in various language tasks. However, they are prone to spurious correlations, and often generate illusory information. In real-world applications, PLMs should justify decisions with formalized, coherent reasoning chains, but this challenge remains under-explored. Cognitive psychology theorizes that humans are capable of utilizing fast and intuitive *heuristic* thinking to make decisions based on past experience, then rationalizing the decisions through slower and deliberative *analytic* reasoning. We incorporate these interlinked dual processes in fine-tuning and in-context learning with PLMs, applying them to two language understanding tasks that require coherent physical commonsense reasoning. We show that our proposed Heuristic-Analytic Reasoning (HAR) strategies drastically improve the coherence of rationalizations for model decisions, yielding state-of-the-art results on Tiered Reasoning for Intuitive Physics (TRIP). We also find that this improved coherence is a direct result of more faithful attention to relevant language context in each step of reasoning. Our findings suggest that human-like reasoning strategies can effectively improve the coherence and reliability of PLM reasoning.

## 1 Introduction

Pre-trained language models (PLMs) have recently shown strong performance on an increasing variety of challenging reasoning problems (Radford et al., 2018, 2019; Devlin et al., 2019; Brown et al., 2020; OpenAI, 2023). While their capabilities are impressive, their tendency to make artificial mistakes in reasoning, including exploiting spurious correlations (Schwartz et al., 2017; Gururangan et al., 2018) and hallucinating factual information (Dziri et al., 2022; Ji et al., 2023) in order to reach conclusions, makes them difficult to rely on in practice. This possibility of incoherence in reasoning with PLMs creates a demand for more reliable, logical, and transparent reasoning strategies compatible with differentiable architectures like PLMs (Marcus, 2020; LeCun, 2022).

Meanwhile, in theories of cognitive psychology, drawing conclusions in reasoning problems and coherently rationalizing them have long been thought to come from dual processes of human cognition (Wason and Evans, 1974; Evans and Wason, 1976; Evans, 1984, 2003; Tsujii and Watanabe, 2009; Evans, 2011): fast, associative *heuristic* thinking based on experience, and slower, deliberative *analytic* thinking, which requires more working memory. Specifically, prior work theorizes that heuristic processes enable us to extract the most relevant information from the context and provide quick intuition for decisions, which can then inform analytic processes that operate on this information to perform inference and rationalize when needed (Evans, 1984, 2010; Kahneman, 2011).

In this work, inspired by the synergy between these dual processes in humans, we propose analogous heuristic-analytic reasoning (HAR) strategies for PLMs, which bootstrap lower-level (analytic) rationalization from higher-level (heuristic) decision-making. Using physical commonsense as an example, we implemented HAR for both fine-tuning and in-context learning. Through evaluation on two benchmarks, Tiered Reasoning for Intuitive Physics (TRIP; Storks et al., 2021) and a novel reframing of ProPara (Dalvi et al., 2018) requiring multiple levels of reasoning, we find that HAR consistently enables more coherent reasoning, achieving state-of-the-art results on coherence metrics in TRIP. More importantly, our analysis shows this improved coherence in in-context learning is enabled by more faithful and transparent rationalization through stronger attention to relevant language context in reasoning. This demonstrates that similar to humans, PLMs' reasoning is strengthened by linking heuristic and analytic processes, and such

---

*Authors contributed equally to this work.

strategies may be applied in other reasoning tasks for more trustworthy decision-making.

## 2 Related Work

**Dual process theory in AI.** Dual process theories of cognitive psychology have recently attracted interest in some areas of AI. Anthony et al. (2017) apply them to augment reinforcement learning algorithms with deliberative planning of policies through tree search. Ganapini et al. (2022) combine them for more efficient navigation with AI agents that evolve from slow to fast decision-making while navigating. Similarly inspired by dual process theories, Hua and Zhang (2022) apply logical reasoning over representation learning for more accurate commonsense knowledge base completion. Lee et al. (2023) use dual-process inspired associative selection combined with evidence generation to perform question answering on scientific articles. Booch et al. (2021) propose additional research questions and directions around the application of dual process theories of human cognition in AI. Unlike these past works, we apply dual process theories of human cognition in coherent physical commonsense reasoning with PLMs, both through fine-tuning and in-context learning.

**Reasoning with PLMs.** Prior work has studied fine-tuning PLMs on various reasoning tasks, including mathematical reasoning (Geva et al., 2020), temporal reasoning (Zhou et al., 2021), and commonsense reasoning (Rajani et al., 2019; Sap et al., 2019), mostly through data-driven approaches to strengthen PLMs' reasoning abilities. Recent work has attempted to build in more explicit coherent reasoning structures (Storks et al., 2021; Ma et al., 2022; Richardson et al., 2022; Li et al., 2023), but these approaches do not take advantage of how higher-level tasks may inform lower-level tasks, rather they reason from the bottom up over various representations of the world state, or jointly optimize all reasoning steps without dependency. Unlike these works, we take inspiration from human reasoning to explore the role of sequential heuristic and analytic processes in fine-tuning and in-context learning, bootstrapping low-level commonsense rationalization from higher-level decisions.

With the introduction of GPT-3 (Brown et al., 2020), in-context learning became a common way to apply PLMs to new tasks without in-domain gradient-based training, where one prompts the PLM with helpful task-specific knowledge or even full demonstrations at inference time before asking it to solve a task. A number of works found applications of in-context learning in PLMs for complex reasoning tasks (Dalvi et al., 2021; Tafjord et al., 2021; Spiliopoulou et al., 2022). Among these works, significant improvements came from inserting or generating free-text reasoning chains in prompts to support task predictions (Wei et al., 2022; Kojima et al., 2022). These findings sparked the exploration of many different in-context learning and sequential prompting approaches to strengthen reasoning in PLMs and tackle various tasks (Kazemi et al., 2023; Nye et al., 2022; Zelikman et al., 2022; Wu et al., 2022; Yao et al., 2023; Long, 2023; Wang et al., 2023). These methods usually rely on an assumption that by decomposing a high-level task into many low-level sub-tasks, the model can solve the low-level sub-tasks easily, which helps it achieve better performance on the high-level task. However, in complex cases like commonsense reasoning, even lower-level sub-tasks are hard to solve due to the requirement of retrieving and incorporating knowledge beyond the text. As such, our heuristic-analytic in-context learning approach instead uses higher-level decisions to refine the generation of low-level commonsense knowledge from PLMs.

## 3 Coherent Commonsense Reasoning

We study the problem of *coherent* commonsense reasoning in natural language understanding. Here, coherent reasoning refers to the ability of a language understanding system to justify a high-level decision on a reasoning task with valid low-level evidence from or beyond the language context that supports the decision (Storks and Chai, 2021; Storks et al., 2021). We specifically focus on the domain of *physical commonsense*: humans' intuitive set of rules and concepts that explain how the world physically works. Since humans build their physical commonsense intuition from a young age (Bliss, 2008; Lake et al., 2017), this knowledge may be considered obvious to most humans (Forbes and Choi, 2017), creating a challenging problem for PLMs which cannot directly interact with the physical world, and may not encounter much of this knowledge in pre-training. While humans rarely communicate about physical commonsense in language or carefully rationalize their intuitions in terms of low-level physical states, it is essential for AI systems to do so for safety, transparency, and

Figure 1: TRIP (left; Storks et al., 2021) and proposed Tiered-ProPara (right) tasks for coherent physical commonsense reasoning. Each task requires multiple levels of reasoning from surface-level story and sentence selection and commonsense physical state prediction. While *accuracy* only evaluates the ability to perform the highest-level task, *consistency* and *verifiability* are used to evaluate lower-level steps and judge the coherence of reasoning.

trustworthiness.

In the context of coherence, we define physical commonsense reasoning (PCR) as the application of this complex background knowledge to understand language describing physical situations and procedures. PCR may include inferring low-level, concrete physical states of participants in text (Dalvi et al., 2018; Zellers et al., 2021), or reasoning over them to perform higher-level tasks, e.g., answering questions about procedures (Bisk et al., 2020; Storks et al., 2021). We focus on two benchmarks that include both types of PCR tasks in multi-step chains enabling evaluation of coherence: Tiered Reasoning for Intuitive Physics (TRIP; Storks et al., 2021) and a new reframing of ProPara (Dalvi et al., 2018) called Tiered-ProPara, examples of which are shown in Figure 1.

## 3.1 TRIP

Tiered Reasoning for Intuitive Physics (TRIP) is a benchmark for measuring coherence of PCR, where high-level predictions over a procedural text must be justified with low-level physical commonsense evidence, i.e., physical states of entities (Storks et al., 2021). As shown in Figure 1, given two stories (A and B), a system predicts which story is more plausible; a system's ability to perform this end task can be measured by traditional classification *accuracy*. It then identifies which two sentences are conflicting in the implausible story, and *consistency* measures how often a system can support an accurate plausibility prediction with correct conflicting sentences. Lastly, the system predicts precondition and effect physical states over

20 physical attributes to justify the identified conflict, and *verifiability* measures how often a system can produce a fully correct reasoning chain including these physical states,[1] conflicting sentences, and story plausibility. Consistency and verifiability both require an accurate story plausibility prediction as a prerequisite, but additionally evaluate the coherence of PLMs' reasoning to support the prediction. To demonstrate a deeper understanding, an AI system should not just achieve high accuracy on TRIP, but comparably high consistency and verifiability. If consistency and verifiability are lower than accuracy, this indicates incoherent reasoning, as the PLM is making correct predictions on the end task without valid underlying support.

Two recent state-of-the-art approaches for TRIP are Coalescing Global and Local Information (CGLI; Ma et al., 2022) and Breakpoint Transformer (Richardson et al., 2022). While they achieve up to near-perfect testing accuracy, consistency and verifiability still lag behind, reaching a respective maximum of 77.3% and 32.4%. Therefore, coherent PCR remains a difficult problem.

## 3.2 Tiered-ProPara

ProPara is a dataset of texts about scientific processes annotated with the dynamic existence and location of entities throughout the processes (Dalvi et al., 2018). While ProPara originally focused on the low-level task of predicting the states of entities before and after each sentence of passages, we pro-

---

[1] Verifiability requires that at least one non-default physical state label is predicted within each conflicting sentence, and all such predicted states are correct.

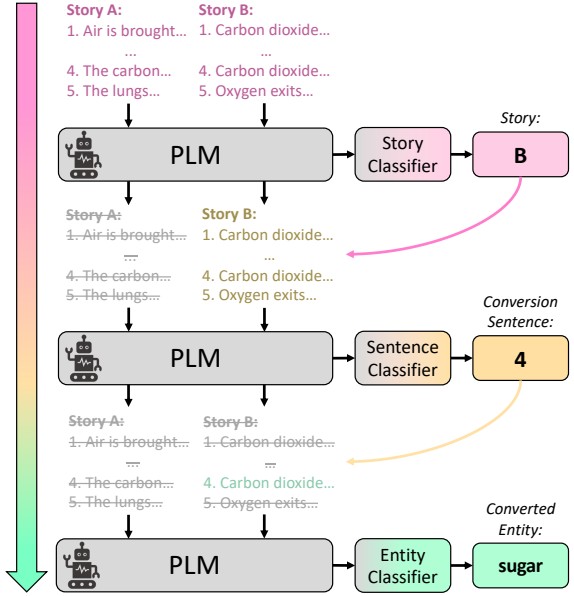

Figure 2: Heuristic-analytic reasoning for fine-tuning PLMs, where the language context is iteratively refined using classification predictions during training and inference. In Tiered-ProPara, after the PLM is used to classify which story contains a conversion, the other story is deleted from the model inputs. After classifying which sentence describes the conversion, other sentences are deleted. Lastly, the resulting entity after the conversion is identified.

pose Tiered-ProPara, a novel reframing of the task which requires multiple levels of reasoning. As shown in Figure 1, in this version of the task, a system is presented with two passages from ProPara with shared entities, and asked in which story is a particular type of entity, e.g., *carbon dioxide*, converted into another entity. In addition to this, the system must identify the sentence in that story in which the conversion occurs, and what the entity is converted into. Similarly to TRIP, we can use these two lower-level tasks to evaluate consistency and verifiability of system predictions on the end task of choosing a story. More details on how we generate this data are given in Appendix B.

Motivated by dual process theories for human cognition, we propose *heuristic-analytic reasoning (HAR)* strategies for these tasks for fine-tuning (Section 4) and in-context learning (Section 5) with PLMs. Each strategy allows the PLM to make easier high-level decisions first (e.g., story and sentence selection), then use them to condition low-level rationalization (e.g., physical state prediction), the essence of combined heuristic and analytic processes in human reasoning.

## 4   HAR for PLM Fine-Tuning

Fine-tuning PLMs is one popular approach for adapting them to downstream tasks, applied in recent work toward coherent commonsense reasoning (Ma et al., 2022; Richardson et al., 2022), and suitable for applications with compute or privacy restrictions. As shown in Figure 2, we can build explicit heuristic-analytic reasoning structure into PLM fine-tuning for our target tasks by deleting parts of the language context that are no longer relevant as the model makes predictions for each step of reasoning, both during training and inference. While our approach provides just one example, this reasoning trick can apply to PLM fine-tuning for any multi-step reasoning problem where the most relevant language context to support reasoning changes with each step.

### 4.1   Fine-Tuning Experiments

Next, we introduce our experiments with HAR in fine-tuning PLMs.

**Implementation.**   To implement HAR in fine-tuning, we use the recent state-of-the-art Coalescing Global and Local Information (CGLI) model for TRIP and ProPara (Ma et al., 2022) as a backbone for HAR in fine-tuning. We apply two tricks to better focus the model on relevant information. First, while the original CGLI model takes one text as input, Focused CGLI (FCGLI) takes two texts, enabling the model to consider both together. Additionally, we filtered down training annotations in TRIP for low-level state prediction tasks to only include the most relevant physical states, i.e., those causing conflicts.[2] This allows the model to focus on learning the most important physical states from the high-dimensional hypothesis space. Development and testing data remain unchanged for fairness. FCGLI with heuristic-analytic reasoning (FCGLI-HAR) applies the above iterative deletion strategy to the input context of FCGLI.

**Baselines.**   To measure the advantage of informing low-level reasoning steps with higher-level steps through sequential heuristic and analytic processes in FCGLI-HAR, we use FCGLI, where all reasoning steps are performed jointly without dependency between them, as a baseline. To contextualize TRIP results with past work, we also include RoBERTa (Liu et al., 2019) results from

---

[2]More fine-tuning details provided in Appendix C.

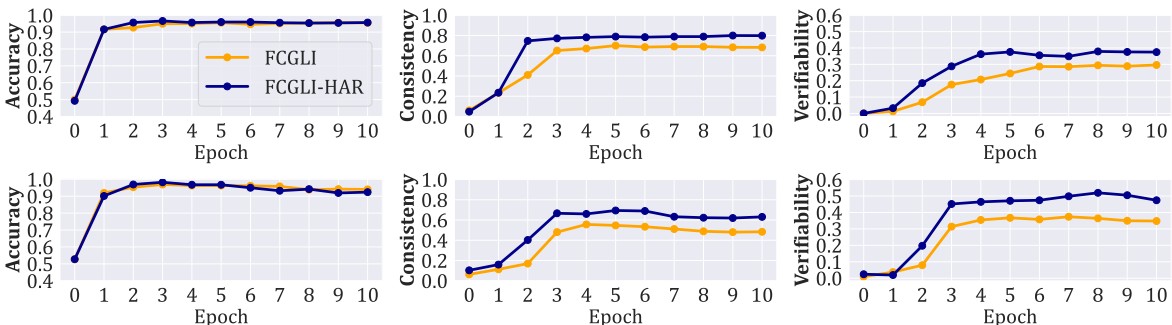

Figure 3: Validation metrics for unstructured FCGLI baseline and FCGLI with heuristic-analytic reasoning (FCGLI-HAR) through epochs of training on TRIP (top) and Tiered-ProPara (bottom).

| Approach | Accuracy | *Consistency* | *Verifiability* |
|---|---|---|---|
| *TRIP* | | | |
| RoBERTa | 72.9 | 19.1 | 9.1 |
| CGLI | 94.1 | **77.3** | 28.0 |
| Breakpoint | 80.6 | 53.8 | 32.4 |
| FCGLI | 93.7 | 66.2 | 33.8 |
| FCGLI-HAR | 94.3 | 75.4 | **41.1** |
| *Tiered-ProPara* | | | |
| FCGLI | 94.5 | 56.7 | 36.2 |
| FCGLI-HAR | 95.1 | **83.6** | **57.4** |

Table 1: TRIP and Tiered-ProPara results for baselines (introduced in Section 4.1), and heuristic-analytic reasoning with fine-tuned PLMs (FCGLI-HAR).

Storks et al. (2021), CGLI (Ma et al., 2022), and Breakpoint Transformer (Richardson et al., 2022).

**Results.**    Results are listed in Table 1.[3] On TRIP, FCGLI-HAR exceeds or achieves comparable performance to baselines on all three metrics, pushing verifiability up to 41.1% and setting a new state-of-the-art result on the most difficult sub-task of TRIP. On Tiered-ProPara, FCGLI-HAR exceeds the FCGLI baseline on all metrics, with consistency and verifiability reaching a respective 83.6% and 57.4%, thus beginning to close the gap between accuracy and these coherence metrics. This shows that *HAR is indeed a promising strategy to improve coherence and dependability of reasoning in PLMs.*

### 4.2   Learning Curves for FCGLI-HAR

We plotted the validation metrics through fine-tuning FCGLI and FCGLI-HAR in Figure 3. Unsurprisingly, we see that consistency and verifiability increase slower than accuracy through training, suggesting that these lower-level objectives are indeed

most difficult to learn. However, FCGLI-HAR converged 1-2 training epochs faster than FCGLI on both datasets, suggesting that HAR may enable more efficient learning of coherent reasoning.

## 5   HAR for PLM In-Context Learning

While HAR can benefit domain-specific applications when integrated into PLM fine-tuning, it requires expensive training on in-domain data that may sacrifice generalizability. To alleviate the limitations of fine-tuning, we also integrated HAR into in-context learning, taking advantage of emergent abilities of large PLMs to perform coherent reasoning through free-form language generation.

Prompting techniques like chain-of-thought (CoT; Wei et al., 2022) can be useful in in-context learning to demonstrate valid reasoning strategies to reach conclusions. However, CoT has traditionally been used to improve performance on complex high-level tasks by breaking them down into simpler low-level steps and reasoning from the bottom up. Meanwhile, the most complex sub-task and bottleneck in our problem is low-level physical state prediction (Storks et al., 2021), which is impossible to further break down, as descriptions of actions directly invoke a world state based on physical commonsense rules of how the world works. Since it would thus be difficult to use traditional CoT to generate a useful explanation to support physical state prediction, we apply HAR through a reverse CoT method where tiered tasks are demonstrated and predicted in a top-down sequence from high-level decisions to low-level rationalization.[4]

In TRIP, as shown in Figure 4, PLMs are conditioned to first predict plausibility, a relatively easy heuristic process. To further refine the relevant con-

---

[3]Statistical significance testing of fine-tuning performance improvements in Appendix A.1.

[4]PLMs are first conditioned with 4 consistent demonstrations of the ICL-HAR strategy from the training set. More details in Appendix D.

Figure 4: Heuristic-analytic reasoning (HAR) for in-context learning with pre-trained language models (PLMs). HAR uses chain-of-thought prompting to bootstrap low-level analytic rationalization (e.g., physical state prediction) from high-level heuristic decision-making (e.g., implausible story and conflicting sentence selection), focusing the PLM's attention to the most relevant context in each reasoning step.

text, the PLM then predicts conflicting sentences, another heuristic judgement that need not be directly based on low-level commonsense knowledge, but may still benefit from being conditioned on the higher-level implausible story prediction. Lastly, the PLM rationalizes these decisions with low-level physical states, an analytic process requiring the integration of external background knowledge about actions and objects. Instead of breaking down physical state prediction further (the typical purpose of CoT), we hypothesize that conditioning this sub-task with higher-level heuristic processes in HAR helps focus the model on the correct context and reason more coherently. We similarly apply HAR in Tiered-ProPara by prompting the PLM to first select a story and sentence in which a particular entity is converted to another, and finally rationalizing these decisions with the name of the resulting

entity after conversion, which requires commonsense understanding of how entities change as a result of various actions and processes.

## 5.1 In-Context Learning Experiments

Next, we introduce our experiments on in-context learning with HAR (ICL-HAR).

**Implementation.** We apply ICL-HAR with InstructGPT[5] (Brown et al., 2020; Ouyang et al., 2022) and LLaMA-65B[6] (Touvron et al., 2023) using greedy decoding. Since LLaMA is limited to a context length of 2048 tokens, while prompts for TRIP include over 3000 tokens to familiarize the model with the physical state classification label space,[7] we apply it to a filtered version of TRIP which only includes instances where annotated states involve only the top-6 most frequent physical precondition-effect pairs.

**Unstructured baseline.** We propose a more traditional, unstructured in-context learning (ICL-U) baseline to help measure the advantage of HAR strategies applied in in-context learning. Instead of prompting PLMs to predict all 3 reasoning steps in sequence, we extract each step of the task through separate but comparable prompts.[8] The PLM is provided both input stories for each task, and given 4 task-specific demonstrations comparable to ICL-HAR. We then combine the 3 extracted predictions on each testing example into one reasoning chain to calculate evaluation metrics on them.

**Traditional chain-of-thought baseline.** Despite the anticipated limitations discussed above, we augment the ICL-U baseline with traditional CoT for comparison, creating an additional in-context learning with CoT (ICL-CoT) baseline. Specifically, we prompt InstructGPT[9] with "let's think step by step about ⟨sub-task⟩" (Kojima et al., 2022) to generate a free-text explanation for each separate reasoning step. We then append these explanations to their respective prompts before eliciting predictions from

---

[5]Specifically, we use the `text-davinci-002` version through an Azure OpenAI deployment.

[6]Specifically, we use a HuggingFace (Wolf et al., 2020) compatible version available at https://huggingface.co/decapoda-research/llama-65b-hf at the time of writing.

[7]See Appendix D.1 for more information.

[8]Example prompts provided in Appendix D.2.

[9]Based on preliminary experiments with both PLMs, we expected InstructGPT to generate more reasonable explanations on the in-context demonstration examples, so we used its explanations in prompting both InstructGPT and LLaMA.

the models. Unlike HAR, which enforces a top-down chain-of-thought, this traditional application of CoT allows the PLM to attempt to break down each step before making a prediction for it.

**Results.** As shown in Table 2, we observe sharp performance improvements[10] from ICL-HAR over the baselines, particularly in the coherence metrics of consistency and verifiability, where our proposed strategy primarily comes into play. Meanwhile, compared to both baselines, HAR improves InstructGPT consistency on TRIP from 40.7% up to 47.9%, and verifiability from a maximum of 10.8% up to 23.9%, over a 100% improvement on the latter. LLaMA sees similar improvements, especially in verifiability. On Tiered-ProPara, InstructGPT's consistency improves from 19.2% to 31.5%, and verifiability improves from 7.5% to 20.7%, nearly a 200% improvement on the latter. LLaMA again sees similar improvements. These results demonstrate that compared to common approaches for in-context learning with PLMs, *human-inspired heuristic-analytic reasoning can significantly improve coherence and reduce hallucination.*

As expected, *traditional chain-of-thought in the ICL-CoT baseline brought only marginal improvements in most cases*, especially for verifiability, as physical state prediction (the bottleneck in coherent physical commonsense) cannot be further decomposed. Instead, we saw that the free-form explanations generated by InstructGPT for physical state prediction typically repeated specific sentences and actions from the story, which introduced no new information. ICL-HAR reveals a possible new use case for chain-of-thought-style prompting: refining PLM attention to the most relevant language context. We investigate this further in Section 5.2.

Interestingly, *HAR not only brought vast improvements on verifiability metrics, but also some improvements in consistency*. In other words, the mid-level sentence selection tasks benefited slightly from being conditioned on higher-level story selection tasks, despite being considered heuristic processes that simply refine the context. This may suggest that rather than belonging to separate dual processes, these consecutive steps of reasoning may fall along a spectrum from heuristic to analytic processes in a recursive manner in PLMs.

We also observed that LLaMA's accuracy decreased with ICL-HAR. As discussed in Sec-

---

*InstructGPT*

| Approach | TRIP | | | Tiered-ProPara | | |
| --- | --- | --- | --- | --- | --- | --- |
| | Acc. | *Cons.* | *Ver.* | Acc. | *Cons.* | *Ver.* |
| ICL-U | 70.9 | 40.7 | 7.1 | 54.9 | 17.4 | 5.2 |
| ICL-CoT | 75.0 | 40.7 | 10.8 | 50.7 | 19.2 | 7.5 |
| ICL-HAR | 72.6 | **47.9** | **23.9** | 54.9 | **31.5** | **20.7** |

*LLaMA*

| Approach | TRIP | | | Tiered-ProPara | | |
| --- | --- | --- | --- | --- | --- | --- |
| | Acc. | *Cons.* | *Ver.* | Acc. | *Cons.* | *Ver.* |
| ICL-U | 70.4 | 42.3 | 14.8 | 51.2 | 3.8 | 1.4 |
| ICL-CoT | 74.6 | 42.3 | 19.7 | 57.3 | 9.4 | 4.2 |
| ICL-HAR | 55.6 | **44.4** | **35.2** | 41.8 | **17.8** | **13.1** |

Table 2: Accuracy, consistency, and verifiability percentages for in-context learning with heuristic-analytic reasoning (ICL-HAR) in PLMs, compared to an unstructured in-context learning (ICL-U) baseline that tackles reasoning steps through separate focused prompts.

tion 3.1, it is not ideal for accuracy to far exceed consistency and verifiability, each of which require end-task predictions to be accurate as a prerequisite, as this indicates incoherent reasoning with insufficient support for the end-task prediction. Therefore, *drops in accuracy with HAR are not necessarily problematic*. Nonetheless, we believe it occurs due to the smaller complexity of LLaMA (65B), making it more sensitive to long prompts and generations. A quick fix could be to sequentially prompt the PLM multiple times with shorter prompts for each reasoning step, where the prompt for each step is informed by the previous step's prediction. We explore this approach in Appendix A.2, and indeed find the accuracy does not drop with HAR.

## 5.2 Faithful Attention in ICL-HAR

To explore possible reasons for why HAR strengthens PLMs' reasoning, we last compare and examine models' attention weights when generating language in our in-context learning experiments, where the model has access to the entire input language context throughout inference. Earlier in this section, we hypothesized that ICL-HAR enables the model to focus in on key parts of the context to make decisions and rationalize them more faithfully, similar to its role in human reasoning. For example, as shown in Figure 4, after a PLM identifies the implausible story in TRIP, it should attend more to that story when identifying conflicting sentences. After identifying these sentences, it should attend more to those sentences when generating physical commonsense evidence.

In order to validate this hypothesis, we ag-

---

[10]Statistical significance testing of in-context learning performance improvements in Appendix A.1.

gregate and normalize transformer self-attention weights for each story or sentence within the input prompt,[11] then use the ground truth reasoning chains for TRIP and Tiered-ProPara to evaluate the faithfulness of them. To our knowledge, prior work has not studied attention weights in this way for in-context learning with PLMs, so we hope to yield new insights on the nature of reasoning with them. We next introduce the evaluation criteria used for attention weights, then the results of the analysis.

### 5.2.1 Attention Evaluation Criteria

We propose two kinds of measures to capture the faithfulness of attention and its relationship with coherence in TRIP and Tiered-ProPara: attentional ratios and attentional precision and recall.

**Attentional ratios.** To evaluate the faithfulness of PLMs' attention, we can compare the attention weights for the correct segments of language context (i.e., stories or sentences) versus others through an *attentional ratio*. In both TRIP and ProPara, the model must first identify one of two stories containing some physical phenomenon before identifying which sentence(s) in that story contain it (*sentence selection step*). In TRIP, in cases where it correctly identifies the implausible story, we can calculate the attentional ratio for sentence selection by taking the ratio of the mean attention weight of the implausible story (i.e., where the PLM must attend to identify conflicting sentences) to that of the plausible story. Similarly, in Tiered-ProPara, when the model correctly identifies which story contains an entity conversion, we calculate the ratio of the mean attention weight for the story containing an entity conversion to that of the story that does not.

When the model correctly identifies which sentence(s) contain a phenomenon (i.e., a plausibility conflict or entity conversion), the model must lastly generate physical commonsense knowledge for those sentences to rationalize its decisions (*physical state prediction step*). In TRIP, we calculate the attentional ratio for physical state prediction by taking the ratio of the mean attention weight for conflicting sentences (i.e., sentences from which physical states must be predicted) to that of all other sentences. In Tiered-ProPara, we similarly calculate the ratio between the mean attention weights for the conversion sentence and other sentences.

Together, these ratios can provide a sense of how strongly the PLM is attending to the relevant

language context to produce each level of the reasoning chain. We expect that higher ratios indicate more faithful rationalizations from the model.

**Attentional precision and recall.** Beyond the faithfulness of attention, we would like to understand how faithful attention relates to coherent reasoning, i.e., the PLM's predicted sentence(s) and physical state(s) to rationalize which story it chose. For each of these reasoning steps, there are four possible combinations of faithfulness of model attention and correctness of its predictions:[12]

1. Attends to the *correct* context, and generates a *correct* prediction (**true positive**)
2. Attends to the *correct* context, but generates an *incorrect* prediction (**false positive**)
3. Attends to the *incorrect* context, and generates an *incorrect* prediction (**true negative**)
4. Attends to the *incorrect* context, but generates an *correct* prediction (**false negative**)

We can calculate the precision and recall of attention to measure how the correctness of attended language context correlates with correctness of these model predictions (and thus coherence of reasoning). Given a set of evaluation examples, we define *attentional precision* as the number of true positives divided by all positives, representing how often the PLM is correct given faithful attention. We define *attentional recall* as the number of true positives divided by the sum of true positives and false negatives, representing how often the PLM attends faithfully given a correct prediction. Together, these metrics can provide an impression of the connection between faithful attention and coherent reasoning under different prompting strategies.

### 5.2.2 Attention Analysis Results

To understand why HAR improves coherence so significantly, we compare PLM self-attention distributions as a reasoning chain is generated in the in-context learning setting. As this analysis requires access to PLM internals, we can only use open-source models like LLaMA here. Following findings from prior work that the middle layers of transformer-based language representations contain the most transferable semantic information (Peters et al., 2018; Tenney et al., 2019), we extract self-attention weights from the center-most 20 layers of the transformer backbone in LLaMA.

---

[11]More details in Appendix E.1.

[12]Attention faithfulness classified by a threshold. More details and supporting examples listed in Appendix E.2.

| *Sentence Selection Step* | | | | | | |
|---|---|---|---|---|---|---|
| | **TRIP** | | | **Tiered-ProPara** | | |
| **Approach** | Ratio | Prec. | Rec. | Ratio | Prec. | Rec. |
| ICL-U | 0.96 | 42.6 | 39.6 | 0.90 | 14.8 | 30.6 |
| ICL-HAR | **1.07** | **75.2** | **48.7** | **1.80** | **51.1** | **58.2** |

| *Physical State Prediction Step* | | | | | | |
|---|---|---|---|---|---|---|
| | **TRIP** | | | **Tiered-ProPara** | | |
| **Approach** | Ratio | Prec. | Rec. | Ratio | Prec. | Rec. |
| ICL-U | 1.23 | 43.0 | 35.4 | 1.21 | 14.6 | 25.9 |
| ICL-HAR | **1.95** | **79.8** | **98.2** | **2.20** | **72.1** | **83.3** |

Table 3: Attentional ratio, average precision (%), and average recall (%) for LLaMA baseline and HAR strategy, during different physical commonsense reasoning steps. Precision and recall averaged across several attention thresholds, as outlined in Appendix E.2.

Story A: 41.0%
1. Coal is heated in the boiler.
2. The water tank over the boiler is heated.
3. Creates steam.
4. The steam is funneled to the piston.
5. Piston uses the steam as energy.
6. The piston causes the crankshaft to move.
Story B: 59.0%
1. Plates on the Earth's crust move slowly past each other.
2. As the plates move, they exert a great force.
3. When the force is large enough, the crust breaks.
4. The stress is released as energy.
5. The energy moves through the Earth in the form of waves.
6. We feel the earthquake.

Story A: 16.3%
1. Coal is heated in the boiler.
2. The water tank over the boiler is heated.
3. Creates steam.
4. The steam is funneled to the piston.
5. Piston uses the steam as energy.
6. The piston causes the crankshaft to move.
Story B: 83.7%
1. Plates on the Earth's crust move slowly past each other.
2. As the plates move, they exert a great force.
3. When the force is large enough, the crust breaks.
4. The stress is released as energy.
5. The energy moves through the Earth in the form of waves.
6. We feel the earthquake.

Figure 5: Attention visualization on Tiered-ProPara in selecting which sentence *energy* is converted in, baseline ICL-U (top) vs. ICL-HAR (bottom). Attention averaged across stories and reflected by the intensity of color.

Story A:
1. Tom found he is out of ice cream. 9.0%
2. Tom peeled a hard boiled egg. 5.5%
3. Tom sliced the egg with a knife. 4.6%
4. Tom washed the knife in the sink. 4.4%
5. Tom ate ice cream for dessert. 8.6%
Story B:
1. Tom poured a glass of milk. 10.4%
2. Tom peeled a hard boiled egg. 25.4%
3. Tom sliced the egg with a knife. 3.3%
4. Tom washed the knife in the sink. 16.2%
5. Tom ate ice cream for dessert. 12.5%

Story A:
1. Tom found he is out of ice cream. 21.3%
2. Tom peeled a hard boiled egg. 7.1%
3. Tom sliced the egg with a knife. 5.3%
4. Tom washed the knife in the sink. 4.4%
5. Tom ate ice cream for dessert. 15.4%
Story B:
1. Tom poured a glass of milk. 7.2%
2. Tom peeled a hard boiled egg. 8.2%
3. Tom sliced the egg with a knife. 2.4%
4. Tom washed the knife in the sink. 20.8%
5. Tom ate ice cream for dessert. 7.9%

Figure 6: Sentence-wise attention visualization on TRIP in state change prediction for baseline ICL-U (left) vs. ICL-HAR (right). Attention averaged across sentences.

As shown in Table 3, ICL-HAR sharply exceeds the unstructured ICL-U baseline across all attentional metrics, both when selecting sentences (i.e., conflicting sentences or sentences with con-

versions) and predicting states of entities. While observed ratios show that *ICL-HAR has more faithful attention to relevant parts of the context*, the high values of attentional precision (up to 80%) and recall (up to 98%) show that *faithful attention and coherent reasoning go hand-in-hand*; faithful attention in ICL-HAR is likely to bring coherent reasoning, and vice-versa. This demonstrates that HAR enables more trustworthy reasoning in PLMs.

Lastly, we present example visualizations of self-attention patterns (Yang and Zhang, 2018) for ICL-HAR compared to the ICL-U baseline. In Figure 5, we see that in the sentence selection step in Tiered-ProPara, LLaMA had higher average attention on the story containing a conversion of *energy* under ICL-HAR than the baseline. Similarly, in Figure 6, we see that under ICL-HAR, LLaMA paid more attention to the conflicting sentences in the physical state prediction step in TRIP, whereas the baseline had high attention on irrelevant sentences. This shows how HAR can help focus PLMs on the most relevant language context at each step of reasoning. More visualizations are provided in Appendix A.3.

## 6 Conclusion and Future Work

In this work, we took inspiration from the synergy between heuristic and analytic processes in human reasoning to explore how high-level decision-making tasks in commonsense reasoning can condition and drive lower-level rationalization tasks supporting them in PLMs. We proposed two general strategies to integrate heuristic-analytic reasoning (HAR) into PLM fine-tuning and in-context learning, and found that HAR sharply improved reasoning coherence, outperforming competitive baselines on two benchmark tasks. In fine-tuning, we saw that HAR enabled PLMs to learn to reason not only more coherently, but also faster. Meanwhile, in in-context learning, we found that improvements were enabled by more faithful attention to the language context within each step of reasoning, shedding light on the nature of incoherence in language generation. While this human-inspired approach shows promising strides toward more trustworthy reasoning from PLMs, future work should continue to dive deeper into cognitively motivated strategies to further strengthen coherent reasoning in AI systems and improve human-machine alignment.[13]

---

[13]Source code to reproduce results at https://github.com/sled-group/Heuristic-Analytic-Reasoning.

## Acknowledgements

This work was supported by LG AI Research. We would like to thank the anonymous reviewers for their valuable comments and suggestions.

## Limitations

**Cascading errors in HAR.** In all presented approaches, model outputs may be influenced by decisions on higher-level tasks, whether explicitly through modifying language context to remove information based on previous decisions, or more indirectly from in-context learning. As the model cannot backtrack from incorrect decisions, this can cause cascading errors when the model makes a mistake. For example, if the model predicts the incorrect conflicting sentences, there is no way for it to recover in lower-level tasks. While our goal was to demonstrate that heuristic decisions can be used to improve performance on analytic rationalizations in PLMs, more robust, non-greedy decoding strategies may help alleviate this problem. For example, a beam search or more recent work in tree search for reasoning in in-context learning could be applied (Yao et al., 2023; Long, 2023) to generate multiple candidate reasoning chains and choose one based on language model likelihoods. In fact, such strategies could strengthen all coherent reasoning approaches studied in this work, including those from prior work. As this effort is thus orthogonal to our goal and would increase the number of variables in our study, we chose not to explore these approaches here and avoid distracting from our key message. Furthermore, this requires the ability to systematically extract and compare all intermediate reasoning steps, which is possible due to the dense annotations in TRIP and ProPara, but may not be possible in all applications.

**Generalizability of approach.** A possible limitation of this work is that our approach is suited to coherent reasoning tasks where a high-level decision needs to be supported by lower-level reasoning which depends on small segments of the language context containing key information. We argue that problems requiring heuristic-analytic reasoning are actually quite prevalent in the real world; for example, when debugging code, an experienced developer will localize a bug to a small segment of the code before closely tracing through it to solve the problem.

Furthermore, coherent reasoning is an impor-

tant and under-explored task, evidenced by current PLMs' tendency to perform incoherent reasoning and hallucinate, making them difficult to use for reasoning in practice. Providing detailed evidence for conclusions in reasoning is especially important in physical commonsense, as agents must have a deep understanding of the possible implications of actions on the environment around them in order to safely collaborate with humans. Further, despite our focused scope, our work proposes attention-based coherence metrics upon existing strategies for evaluating reasoning in PLMs. This enables future work to dive deeper into this problem across domains, and better characterize the domains where various strategies for applying PLMs are helpful. Findings from such efforts may someday lead to the integration of such reasoning strategies into LM pre-training from the ground up to improve their reliability.

## Ethics Statement

This works deals with applications of large pre-trained language models (PLMs), which are prone to bias in their training data. Before deploying PLMs for real-world reasoning applications that interface with users, efforts must be taken to mitigate such bias. Bender et al. (2021) provide an overview of such possible dangers of PLMs and recommend some solutions. On the other hand, by emphasizing consistent and verifiable reasoning, this work brings a potential positive impact toward trustworthiness and transparency in interacting with these systems, especially for embodied AI applications where coherent physical commonsense reasoning is necessary for safety of human users.

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

## A Supplementary Results

Here, we include several supplementary results that could not fit in the main body of our paper, but shed more light on PLM performance under HAR-inspired strategies.

### A.1 Statistical Significance Testing

While the performance gain in consistency and verifiability from our HAR strategies over baseline approaches was quite large in most cases, we performed McNemar's tests to measure the statistical significance of these gains (McNemar, 1947). We found that in the TRIP and Tiered-ProPara results presented in Table 1, the differences in consistency and verifiability between FCGLI-HAR and the FCGLI baseline were statistically significant ($p < 0.05$). This was also true for the differences in consistency and verifiability on both tasks between ICL-HAR and both ICL-U and ICL-CoT in Table 2, except for LLaMA on TRIP, where the consistency gain from ICL-HAR over either baseline was not significant. As expected, HAR indeed brings about statistically significant performance gains in verifiability of physical commonsense reasoning, while sometimes also significantly improving consistency.

### A.2 HAR for Multi-Prompt In-Context Learning

On top of the chain-of-thought (CoT) implementation of HAR for in-context learning presented in the paper, we experimented with a more strict form which chained multiple prompts together, one for each step of the reasoning tasks. This adds explicit structure that may make it more dependable. As shown in Figure 7, each successive prompt is generated based on the previous higher-level prediction. In TRIP and Tiered-ProPara, PLMs must first select one of two stories in which some phenomenon occurs (i.e., implausibility or conversion of entities). Then, the chosen story is used in a separate prompt to that PLM and predict the sentence(s) where the phenomenon occurs. Lastly, given predicted sentence(s), PLMs should predict the specific states underlying the phenomenon. While similar to the approach presented in the paper, this approach enables us to completely remove irrelevant information from the context at each step of reasoning and rationalization, so that the model can focus only on the correct parts of the language context. While this restriction may help PLMs ignore irrelevant con-

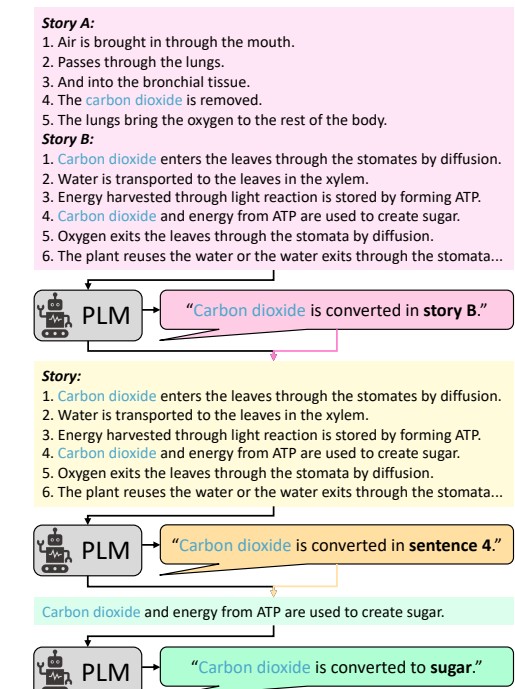

Figure 7: Heuristic-analytic reasoning with prompt chaining for in-context learning with PLMs on physical commonsense rationalization (PCICL-HAR). On Tiered-ProPara, the PLM will first decide which story a conversion of an entity occurs in, then this will be used to refine the language prompt before asking the PLM which sentence the conversion occurs in. Lastly, the chosen sentence will be used to predict the resulting entity after conversion.

text, this structure limits the general applicabilty of this approach compared to the more flexible chain-of-thought approach presented in the paper.

As shown in Tables 4 and 5, prompt chaining for in-context learning with HAR (PCICL-HAR) also brings performance improvements over the ICR-U baseline in both TRIP and ProPara. In Tiered-ProPara, PCICL-HAR slightly exceeds HAR with chain-of-thought (ICL-HAR) in consistency when applied to both InstructGPT and LLaMA (up to 36.2% and 21.6% respectively), and in verifiability with LLaMA (up to 17.4%). This suggests that in some cases, more explicitly structured HAR can be beneficial.

### A.3 Attention Weight Visualizations

We present additional visualizations of self-attention patterns under in-context learning with heuristic-analytic reasoning (ICL-HAR), compared

---

[14]As mentioned in Section 5.1, LLaMA is evaluated on a subset of TRIP, so in-context learning results on different PLMs are not directly comparable.

| InstructGPT | | | |
|---|---|---|---|
| **Approach** | **Accuracy** | *Consistency* | *Verifiability* |
| ICL-U | 70.9 | 40.7 | 7.1 |
| ICL-CoT | 75.0 | 40.7 | 10.8 |
| ICL-HAR | 72.6 | **47.9** | **23.9** |
| PCICL-HAR | 70.4 | 39.6 | 12.8 |
| LLaMA | | | |
| **Approach** | **Accuracy** | *Consistency* | *Verifiability* |
| ICL-U | 70.4 | 42.3 | 14.8 |
| ICL-CoT | 74.6 | 42.3 | 19.7 |
| ICL-HAR | 55.6 | **44.4** | **35.2** |
| PCICL-HAR | 70.4 | 40.8 | 28.2 |

Table 4: TRIP results of heuristic-analytic reasoning (HAR) strategies in in-context learning with PLMs, including PCICL-HAR, the prompt-chaining alternative to ICL-HAR.[14]

| InstructGPT | | | |
|---|---|---|---|
| **Approach** | **Accuracy** | *Consistency* | *Verifiability* |
| ICL-U | 54.9 | 17.4 | 5.2 |
| ICL-CoT | 50.7 | 19.2 | 7.5 |
| ICL-HAR | 54.9 | 31.5 | **20.7** |
| PCICL-HAR | 54.5 | **36.2** | 18.8 |
| LLaMA | | | |
| **Approach** | **Accuracy** | *Consistency* | *Verifiability* |
| ICL-U | 51.2 | 3.8 | 1.4 |
| ICL-CoT | 57.3 | 9.4 | 4.2 |
| ICL-HAR | 41.8 | 17.8 | 13.1 |
| PCICL-HAR | 51.2 | **21.6** | **17.4** |

Table 5: Tiered-ProPara results of heuristic-analytic reasoning (HAR) strategies in in-context learning with PLMs, including PCICL-HAR, the prompt-chaining alternative to ICL-HAR.

to our unstructured in-context learning (ICL-U) baseline. From Figure 5 and 8, we found that when our HAR method was applied, in second-level sentence selection tasks, the model had higher average attention on the story of interest compared with the baseline. We suspect that this increase in average attention helped increase the performance. Similarly, from Figure 6 and 9, we found that after our HAR method was applied, the model paid higher average attention to the sentence of interest in the third-level physical state prediction tasks.

## A.4 Implicit vs. Explicit Conflicts on TRIP

Plausibility conflicts between physical states in TRIP can take two forms. First, *explicit conflicts* like the one shown in Figure 4 exhibit direct disagreements in physical effect states of a particular entity after one sentence and its precondition states

in a later sentence.[15] Other stories have *implicit*

[15]For example, if *Mary put the cucumber on a plate and tossed the donut in the trash*, then in a later sentence, *Mary ate the donut*, the effect of the former sentence (i.e. *donut is inedible*) conflicts with the precondition of the latter sentence (i.e. *donut is edible*).

Story A: 51.1%
1. The book was on the table.
2. Sarah took the book to the copier.
3. Sarah copied a few pages and used the stapler to keep them together.
4. Sarah could not find her notebook.
5. Sarah put the notebook next to the keyboard.
Story B: 48.9%
1. The book was on the table.
2. Sarah took the book to the copier.
3. Sarah copied a few pages and used the stapler to keep them together.
4. Sarah put the pages inside her notebook.
5. Sarah put the notebook next to the keyboard.

Story A: 61.7%
1. The book was on the table.
2. Sarah took the book to the copier.
3. Sarah copied a few pages and used the stapler to keep them together.
4. Sarah could not find her notebook.
5. Sarah put the notebook next to the keyboard.
Story B: 38.3%
1. The book was on the table.
2. Sarah took the book to the copier.
3. Sarah copied a few pages and used the stapler to keep them together.
4. Sarah put the pages inside her notebook.
5. Sarah put the notebook next to the keyboard.

Figure 8: Story-wise attention visualization on TRIP in sentence-of-conversion detection, ICL-U (top) vs. ICL-HAR (bottom).

Story A:
1. Plants have roots. 10.8%
2. The roots grow out. 3.9%
3. Roots have fibers that are attached to them. 8.6%
4. They attract water. 6.7%
5. They suck up water. 5.1%
6. They absorb the water. 12.1%
Story B:
1. The air is cold. 6.0%
2. Water is in the air. 4.4%
3. The water forms tiny ice crystals. 9.9%
4. The ice crystals collide with each other. 4.3%
5. The ice crystals stick to each other. 3.0%
6. The ice crystals get bigger as more of them stick together. 3.4%
7. The ice crystals get too heavy to be in the air. 3.8%
8. The ice crystals become snowflakes. 9.4%
9. The snow flakes fall to the ground as snow. 8.6%
What happened to water?

Story A:
1. Plants have roots. 2.3%
2. The roots grow out. 1.5%
3. Roots have fibers that are attached to them. 3.4%
4. They attract water. 2.8%
5. They suck up water. 3.2%
6. They absorb the water. 11.4%
Story B:
1. The air is cold. 4.4%
2. Water is in the air. 7.4%
3. The water forms tiny ice crystals. 26.5%
4. The ice crystals collide with each other. 9.9%
5. The ice crystals stick to each other. 4.9%
6. The ice crystals get bigger as more of them stick together. 6.5%
7. The ice crystals get too heavy to be in the air. 4.0%
8. The ice crystals become snowflakes. 6.6%
9. The snow flakes fall to the ground as snow. 5.4%
What happened to water?

Figure 9: Sentence-wise attention visualization on Tiered-ProPara in entity conversion prediction, ICL-U (top) vs. ICL-HAR (bottom).

| *Explicit Conflicts* | | | |
|---|---|---|---|
| **Approach** | **Accuracy** | *Consistency* | *Verifiability* |
| ICL-U | 71.7 | 42.2 | 10.0 |
| ICL-CoT | 73.9 | 41.7 | 16.7 |
| ICL-HAR | 72.2 | **53.3** | **36.1** |
| PCICL-HAR | 70.6 | 47.2 | 17.8 |
| FCGLI | 98.3 | 81.1 | 56.7 |
| FCGLI-HAR | 98.7 | **89.4** | **68.9** |
| *Implicit Conflicts* | | | |
| **Approach** | **Accuracy** | *Consistency* | *Verifiability* |
| ICL-U | 70.2 | 39.2 | 4.1 |
| ICL-CoT | 76.0 | 39.8 | 4.7 |
| ICL-HAR | 73.1 | **42.1** | **11.1** |
| PCICL-HAR | 70.2 | 31.6 | 7.6 |
| FCGLI | 88.7 | 50.2 | 9.3 |
| FCGLI-HAR | 89.5 | **60.3** | **11.3** |

Table 6: TRIP results of heuristic-analytic reasoning (HAR) strategies in in-context learning with Instruct-GPT (top) and fine-tuning FCGLI (bottom) for explicit and implicit plausibility conflicts.

*conflicts* where no such disagreement exists, though the story may violate commonsense expectations.[16]

While our evaluation on LLaMA focused on explicit conflicts to ensure only the top-6 most common physical attributes could be used to rationalize plausibility conflicts, our evaluation on Instruct-GPT included both. As the connection between physical states and plausibility conflicts is unclear in implicit conflicts and our HAR strategies are intended to strengthen lower-level rationalization to support higher-level tasks, we may expect them to be especially beneficial for explicit conflicts. Table 6 includes results for both types of conflicts on the fine-tuning and in-context learning approaches introduced so far. We see that implicit conflicts are more difficult to rationalize across the board, even with traditional chain-of-thought as applied in the ICL-CoT baseline. However, HAR indeed has a much more significant impact on TRIP examples with explicit conflicts, increasing verifiability from 10.0% to 36.1%, compared to an increase from 4.1% to 11.3% on implicit conflicts. This suggests that models' heuristic predictions of which story is plausible can indeed help improve performance on the analytic predictions of conflicting sentences and physical states, the latter of which has a large search space.

---

[16]For example, in one sentence, we may see that *Tom put the soup in the microwave*, which indirectly implies the *soup* should be heated up, then in a later sentence, *Tom ate the cold soup*.

## B Tiered-ProPara Generation Details

Tiered-ProPara was generated by a simple matching process over the original dataset. We first re-split the dataset to distribute more stories into the testing and development sets from training set. Then, we applied a pairwise matching process to select two stories. Specifically, we enforce that the selected entity appears in both stories, but only converts to another entity in one of them. Also, the entity being converted must disappear after the conversion, while the entity converted must appear only after the conversion. The selected entity is provided as a known fact before the inference. Therefore, similar to TRIP, a system needs to predict which story has the conversion of the known entity given two stories, measured by *accuracy*. And *Consistency* is used to measure how often a system predicting the correct story and the sentence that has the conversion. Additionally, *Verifiability* measures how often a system identifies the fully correct reasoning chain, from story and sentence prediction to physical state prediction of understanding what entity has converted to. After our conversion of the dataset, there are 496 training instances, 206 development instances, and 213 testing instances.

## C PLM Fine-Tuning Details

In Section 4, we proposed two fine-tuning approaches powered by CGLI (Ma et al., 2022), which predicts reasoning steps through task-specific layers. These layers are linear projections for the tiered tasks of selecting stories, sentences, and physical states. We preserve the *entity-aware* and *timestep-aware* encodings in CGLI. Specifically, given an *entity* E, we concatenate it with a story pair $S$ to create a prompt $C =$ [CLS] $E$ [SEP] $S$ [SEP]. Following CGLI, $C$ is then mapped with the embedding layer of the language model and summed with the *timestep* embedding. Finally, it is encoded by the language model to create a latent representation for task-specific classifications. We jointly optimize three cross-entropy losses for the story selection step, sentence selection step, and physical state prediction step $(\mathcal{L}_{story} + \mathcal{L}_{sentence} + \mathcal{L}_{state})$.

In TRIP, in order to make our proposed Focused CGLI (FCGLI) models focused on predicting explicit conflicts (defined in Appendix A.4) in stories, we do not optimize the physical state prediction loss for implicit conflicts. As such, the physical state information for each task instance consists of

an entity, an attribute, effect state, and precondition state. The physical state loss is calculated by averaging four losses.

We selected the model for the test set by using the model with the highest validation verifiability for each task. We used a consistent set of hyperparameters across tasks: a learning rate of 5e-6, a maximum of 10 training epochs, and a batch size of 1 (the maximum that could fit in GPU memory). The weight decay is set to be 0.01 for all parameters except for bias and LayerNorm.weight, and we use a warmup scheduler following CGLI. We report the performance by averaging three random runs. All experiments were performed on a single NVIDIA GeForce RTX 3090 Ti graphics card (24GB).

## D  PLM Prompt Details

Here, we include some extra details about prompting PLMs.

### D.1  Automatic Exemplar Generation

We automatized exemplar generation for smoother and consistent experiments on TRIP and Tiered-ProPara.

### D.1.1  TRIP

The lowest level task in TRIP dataset is annotated with symbolic states to represent precondition and effect states over 20 physical states. Therefore, we convert symbolic physical states to natural language. For stories with explicit conflict, we iterate through all combinations of entities and 20 attributes for the 2 conflicting sentences in order to find the conflicting physical states. For some conflicts that have different entity name for the same entity, we used the following algorithm to first find all possible conflicting entity pairs, and then iterate them for the conflicting entity pair which results in maximum cosine similarity ($argmax$) between GLoVe embedding vectors (50-dimensional) (Pennington et al., 2014) and more specifically, the $argmax$ function is defined by:

$$\arg \max_{c \in candidates} \frac{V^c_{GloVe(entity_1)} \cdot V^c_{GloVe(entity_2)}}{\|V^c_{GloVe(entity_1)}\| \|V^c_{GloVe(entity_2)}\|} \quad (1)$$

**Physical state familiarization.**  To prime PLMs for the highly-dimensional physical state classification step of TRIP, all in-context learning experiments are prepended with a list of possible physical

states (e.g. dirty, clean, unpowered, powered) and a 1-shot example for each one.[17] This process is called *familiarization*, and is essential to fully specify the expected outputs for the reasoning task and enable systematic evaluation, as the model will be more likely to predict physical states within this demonstrated space of labels. As Tiered-ProPara's low-level state space consists of entities mentioned in the given text, this is only necessary for TRIP.

**Filtering TRIP to shorten prompts.**  Since LLaMA is limited to a context length of 2048 tokens, we create a filtered version of TRIP which only includes instances where annotated states involve only the top-6 most frequent physical attributes. The familiarization process mentioned above contains $\sim 70\%$ tokens in the full prompt ($\sim 3800$ tokens). Therefore, we first perform a statistical analysis on the dataset and select 6 highest-frequency physical states for effect and precondition, namely *(no longer existent, existent), (broken, functional), (in pieces, whole), (turned off, turned on), (inedible, edible), (unpowered, powered)*. After that, the prompt length is reduced to $\sim 1600$ tokens. As familiarization of physical states is used to familiarize the model with the classification space, we filtered the dataset to stories that only contain explicit conflicts between these high-frequency physical states.

### D.1.2  Tiered-ProPara

The automatic exemplar generation process for Tiered-Propara is similar to the TRIP, but more straightforward. There's no familiarization stage for Tiered-Propara. After two-story prompt, we asked the model with a question: *What happened to [converted entity]?* The answer prompt in the demonstration is composed by filling ground truth labels (story, sentence, and the entity converted to) to the template.

### D.2  Full Prompt Examples

From our in-context learning experiments, we include full example prompts used with InstructGPT for the in-context learning with heuristic-analytic reasoning (ICL-HAR) and unstructured in-context learning (ICL-U) strategies. Figures 10 and 11 show examples for ICL-HAR in TRIP and ProPara,

---

[17]For example, *After Mary sliced the apple, what is the state of the apple? The apple is now in pieces.* and *Before Tom opened the door, what was the state of the door? The door was closed.* may be used to familiarize the model with the physical state classification space.

respectively. Figures 12, 13, and 14 show examples for the ICL-U in TRIP, while Figures 15, 16, and 17 show examples for the ICL-U baseline in ProPara. For the ICL-CoT baseline, we simply append "Let's think step by step..." and zero-shot CoT generated by InstructGPT into the ICL-U prompting demonstrations before final predictions are made for each sub-task. We provide the full zero-shot CoT prompts we used for each sub-task in TRIP and Tiered-ProPara below:

- **TRIP, Story Selection:** Let's think step by step about which story is more plausible.

- **TRIP, Sentence Selection:** Let's think step by step about which sentences are conflicting in one story.

- **TRIP, Physical State Prediction:** Let's think step by step about which physical states are conflicting in two sentences in one story.

- **Tiered-Propara, Story Selection:** Let's think step by step about which story [entity] were converted in.

- **Tiered-Propara, Sentence Selection:** Let's think step by step about which sentence [entity] were converted in one story.

- **Tiered-Propara, Physical State Prediction:** Let's think step by step about what [entity] were converted to in one sentence in one story.

# E Attention Analysis Details

## E.1 Self-Attention Weight Extraction

To enable our attention analysis for soft HAR in in-context learning, we used the *output_attentions* flag in the Transformers (Wolf et al., 2020) library to extract the raw attentions computed during inference with LLaMA. An attention mask is applied to the attentions to remove the attentions associated with the demonstration prompt and special characters, only keeping the attentions associated with the tokens from the test prompt. We then summed up the attentions across each sentence in the test prompt, averaged across a subset of the generated tokens, and normalized by dividing the sum of the attentions. Here, we computed the token subset's average attention on a sentence of the story prompt, and we used it to measure the importance of a sentence to the model's reasoning outcomes. We then used these normalized weights to calculate the evaluation criteria proposed in Section 5.2.

## E.2 Attentional Precision and Recall Details

Attentional precision and recall are calculated by converting normalized attention weights into binary measures of whether attention is correct. To do this, we check whether the average attention weight for the relevant segment of language context (i.e., the appropriate story or sentence(s)) exceeds a threshold. We calculate the average precision and recall over a set of 9 candidate thresholds centered around 0.1 (0.08 - 0.12) with an interval 0.005 because on average, there are about 10 sentences on in a pair of stories, and all sentences' attention are normalized to a sum of 1.

Given this binary measure, we can then classify each PLM output into four combinations over whether its attention is faithful, and whether its intermediate predictions are coherent (i.e., consistent or verifiable): true positive, false positive, true negative, and false negative. We provide some examples here assuming a static threshold of 0.09. In the story-level prediction example from Figure 5, we calculate the average sentence-wise attention in the story containing a conversion (i.e., story B), and compare it to the threshold. These values are $0.590/6 = 0.098$ and $0.837/6 = 0.140$ for ICL-U and ICL-HAR, respectively. Both values of ICL-U and ICL-HAR exceeded the threshold, but ICL-U didn't correctly identify the sentence that contains a conversion, while ICL-HAR did. Therefore, we classify the example from ICL-U as false negative and ICL-HAR as true positive. Similarly, in the physical state detection example from Figure 6, we calculate the average sentence-wise attention on the two conflicting sentences, which are sentences 1 and 5 in story A. These values are $(0.09 + 0.086)/2 = 0.088$ and $(0.213 + 0.154)/2 = 0.184$ for ICL-U and ICL-HAR, resp. Because ICL-HAR exceeded the threshold but ICL-U didn't, and ICL-HAR actually generated the correct response while ICL-U didn't, we classify the example from ICL-U as a true negative and ICL-HAR as a true positive.

**Familiarization Prompts - Physical States (6 of 80):**
Physical state options: powered, edible, whole ...
Before Tom turned on the microwave, what was the state of the microwave? The microwave was powered.
Before Tom ate the cereal, what was the state of the cereal? The cereal was edible.
Before Tom cut the banana into slices, what was the state of the banana? The banana was whole.
...
Physical state options: unpowered, inedible, in pieces ...
After Tom unplugged the microwave, what is the state of the microwave? The microwave is now unpowered.
After Tom threw the cereal in the dustbin, what is the state of the cereal? The cereal is now inedible.
After Tom mashed the banana, what is the state of the banana? The banana is now in pieces.

**Demonstration Prompts - Physical States (1 of 4):**
Story A:
1. Mary went to the fridge.
2. Mary took out a bowl from the fridge.
3. The bowl had a cucumber and a donut in it.
4. Mary put the cucumber on a plate and tossed the donut in the trash.
5. Mary ate the donut.
Story B:
1. Mary went to the fridge.
2. Mary took out a bowl from the fridge.
3. The bowl had a cucumber and a donut in it.
4. Mary put the cucumber on the counter.
5. Mary ate the donut.

Story B is more plausible.
In Story A, sentences 4 and 5 conflict with each other.
For sentence 4:
After Mary put the cucumber on a plate and tossed the donut in the trash, what is the state of the donut?
The donut is now inedible.
For sentence 5:
Before Mary ate the donut, what was the state of the donut? The donut was edible.

**Test Prompt:**
Story A:
1. Ann opened the toolbox and realized that someone took away all of her tools.
2. Ann took out an axe, a pair of scissors, rope and a few other things.
3. Ann put these items in a box and placed them in the trunk of her car.
4. Ann fastened her bicycle to the bicycle rack on her car.
5. Ann went inside the house to get the rest of her gear to put in the car.
Story B:
1. Ann opened the toolbox to get a few things for her camping trip.
2. Ann took out an axe, a pair of scissors, rope and a few other things.
3. Ann put these items in a box and placed them in the trunk of her car.
4. Ann fastened her bicycle to the bicycle rack on her car.
5. Ann went inside the house to get the rest of her gear to put in the car.

**LLM Sample Output:**
*Story B is more plausible.*
*In Story A, sentences 1 and 2 conflict with each other.*
*For sentence 1:*
*After Ann opened the toolbox and realized that someone took away all of her tools, what are the state of the tools? The tools are now no longer existent.*
*For sentence 2:*
*Before Ann took out an axe, a pair of scissors, rope and a few other things, what were the state of the things? The things were existent.*

Figure 10: Sample prompt used for in-context learning with heuristic-analytic reasoning (ICL-HAR) on TRIP. Some redundant sections of the prompt are omitted (indicated with "..." and in section headings).

## Tiered-ProPara - ICL-HAR

**Demonstration Prompts (1 of 4):**
Story A:
1. Wind and water erode rocks.
2. The wind and water pick up sediment that came off the rocks.
3. Sediment is carried to an area by the wind, water, air, or ice.
4. Sediment is deposited on the same spot many times.
5. The sediment piles on top of other sediment for a long time.
6. The sediment near the bottom is compressed by the weight of newer sediment.
7. The sediment becomes sedimentary rock as it is pushed together by the heavy weight.
Story B:
1. Waves hit the coast.
2. Waves have energy that pushes the water and sediment.
3. The sediment can help erode land like sandpaper.
4. Water washes the sediment back.
5. Some sediment is left as sand.
What happened to sediment?
Sediment is converted in story A.
In story A, sediment is converted in sentence 7.
After the sediment becomes sedimentary rock as it is pushed together by the heavy weight, sediment is converted to sedimentary rock.

**Test Prompt:**
Story A:
1. Coal is heated in the boiler.
2. The water tank over the boiler is heated.
3. Creates steam.
4. The steam is funneled to the piston.
5. Piston uses the steam as energy.
6. The piston causes the crankshaft to move.
Story B:
1. Plants obtain water through the soil they are growing in.
2. The plants roots absorb the water from the soil.
3. Transport the water to the parts of the plant where the water is needed.
4. The plant uses the water as part of the photosynthesis process.
5. The plant creates food, called glucose, for itself.
6. The plant uses water to circulate the glucose around the plant to deliver the glucose to where it's needed.
What happened to water?

**LLM Sample Output:**
*Water is converted in story A.*
*In story A, water is converted in sentence 3.*
*After creates steam, water is converted to steam.*

Figure 11: Sample prompt used for in-context learning with heuristic-analytic reasoning (ICL-HAR) on Tiered-ProPara. Some redundant sections of the prompt are omitted (indicated with "..." and in section headings).

## TRIP - ICL-U Baseline - Plausible Story Selection

**Demonstration Prompts (1 of 4):**
Story A:
1. Mary went to the fridge.
2. Mary took out a bowl from the fridge.
3. The bowl had a cucumber and a donut in it.
4. Mary put the cucumber on a plate and tossed the donut in the trash.
5. Mary ate the donut.
Story B:
1. Mary went to the fridge.
2. Mary took out a bowl from the fridge.
3. The bowl had a cucumber and a donut in it.
4. Mary put the cucumber on the counter.
5. Mary ate the donut.
Story B is more plausible.

**Test Prompt:**
Story A:
1. Ann opened the toolbox and realized that someone took away all of her tools.
2. Ann took out an axe, a pair of scissors, rope and a few other things.
3. Ann put these items in a box and placed them in the trunk of her car.
4. Ann fastened her bicycle to the bicycle rack on her car.
5. Ann went inside the house to get the rest of her gear to put in the car.
Story B:
1. Ann opened the toolbox to get a few things for her camping trip.
2. Ann took out an axe, a pair of scissors, rope and a few other things.
3. Ann put these items in a box and placed them in the trunk of her car.
4. Ann fastened her bicycle to the bicycle rack on her car.
5. Ann went inside the house to get the rest of her gear to put in the car.

**LLM Sample Output:**
*Story B is more plausible.*

Figure 12: Sample prompt used for plausible story selection step of unstructured in-context learning (ICL-U) baseline on TRIP. Some redundant sections of the prompt are omitted (indicated with "..." and in section headings).

## TRIP - ICL-U Baseline - Conflicting Sentence Selection

**Demonstration Prompts (1 of 4):**
Story A:
1. Mary went to the fridge.
2. Mary took out a bowl from the fridge.
3. The bowl had a cucumber and a donut in it.
4. Mary put the cucumber on a plate and tossed the donut in the trash.
5. Mary ate the donut.
Story B:
1. Mary went to the fridge.
2. Mary took out a bowl from the fridge.
3. The bowl had a cucumber and a donut in it.
4. Mary put the cucumber on the counter.
5. Mary ate the donut.
Sentences 4 and 5 conflict with each other in story A.

**Test Prompt:**
Story A:
1. Ann opened the toolbox and realized that someone took away all of her tools.
2. Ann took out an axe, a pair of scissors, rope and a few other things.
3. Ann put these items in a box and placed them in the trunk of her car.
4. Ann fastened her bicycle to the bicycle rack on her car.
5. Ann went inside the house to get the rest of her gear to put in the car.
Story B:
1. Ann opened the toolbox to get a few things for her camping trip.
2. Ann took out an axe, a pair of scissors, rope and a few other things.
3. Ann put these items in a box and placed them in the trunk of her car.
4. Ann fastened her bicycle to the bicycle rack on her car.
5. Ann went inside the house to get the rest of her gear to put in the car.

**LLM Sample Output:**
*Sentences 1 and 2 conflict with each other in story A.*

Figure 13: Sample prompt used for conflicting sentence selection step of unstructured in-context learning (ICL-U) baseline on TRIP. Some redundant sections of the prompt are omitted (indicated with "..." and in section headings).

## TRIP - ICL-U Baseline - Physical State Prediction

**Familiarization Prompts - Physical States (6 of 80):**
Physical state options: powered, edible, whole ...
Tom turned on the microwave. Before, what was the state of the microwave? The microwave was powered.
Tom ate the cereal. Before, what was the state of the cereal? The cereal was edible.
Tom cut the banana into slices. Before, what was the state of the banana? The banana was whole.
...
Physical state options: unpowered, inedible, in pieces ...
Tom unplugged the microwave. After, what is the state of the microwave? The microwave is now unpowered.
Tom threw the cereal in the dustbin. After, what is the state of the cereal? The cereal is now inedible.
Tom mashed the banana. After, what is the state of the banana? The banana is now in pieces.

**Demonstration Prompts (1 of 4):**
Story A:
1. Mary went to the fridge.
2. Mary took out a bowl from the fridge.
3. The bowl had a cucumber and a donut in it.
4. Mary put the cucumber on a plate and tossed the donut in the trash.
5. Mary ate the donut.
Story B:
1. Mary went to the fridge.
2. Mary took out a bowl from the fridge.
3. The bowl had a cucumber and a donut in it.
4. Mary put the cucumber on the counter.
5. Mary ate the donut.
After, what is the state of the donut? The donut is now inedible.
Before, what was the state of the donut? The donut was edible.

**Test Prompt:**
Story A:
1. Ann opened the toolbox and realized that someone took away all of her tools.
2. Ann took out an axe, a pair of scissors, rope and a few other things.
3. Ann put these items in a box and placed them in the trunk of her car.
4. Ann fastened her bicycle to the bicycle rack on her car.
5. Ann went inside the house to get the rest of her gear to put in the car.
Story B:
1. Ann opened the toolbox to get a few things for her camping trip.
2. Ann took out an axe, a pair of scissors, rope and a few other things.
3. Ann put these items in a box and placed them in the trunk of her car.
4. Ann fastened her bicycle to the bicycle rack on her car.
5. Ann went inside the house to get the rest of her gear to put in the car.

**LLM Sample Output:**
*After, what are the state of the tools? The tools are now no longer existent.*
*Before, what were the state of the things? The things were existent.*

Figure 14: Sample prompt used for physical state prediction step of unstructured in-context learning (ICL-U) baseline on TRIP. Some redundant sections of the prompt are omitted (indicated with "..." and in section headings).

## Tiered-ProPara - ICL-U Baseline - Conversion Story Selection

**Demonstration Prompts (1 of 4):**
Story A:
1. Wind and water erode rocks.
2. The wind and water pick up sediment that came off the rocks.
3. Sediment is carried to an area by the wind, water, air, or ice.
4. Sediment is deposited on the same spot many times.
5. The sediment piles on top of other sediment for a long time.
6. The sediment near the bottom is compressed by the weight of newer sediment.
7. The sediment becomes sedimentary rock as it is pushed together by the heavy weight.
Story B:
1. Waves hit the coast.
2. Waves have energy that pushes the water and sediment.
3. The sediment can help erode land like sandpaper.
4. Water washes the sediment back.
5. Some sediment is left as sand.
What happened to sediment?
Sediment is converted in story A.

**Test Prompt:**
Story A:
1. Coal is heated in the boiler.
2. The water tank over the boiler is heated.
3. Creates steam.
4. The steam is funneled to the piston.
5. Piston uses the steam as energy.
6. The piston causes the crankshaft to move.
Story B:
1. Plants obtain water through the soil they are growing in.
2. The plants roots absorb the water from the soil.
3. Transport the water to the parts of the plant where the water is needed.
4. The plant uses the water as part of the photosynthesis process.
5. The plant creates food, called glucose, for itself.
6. The plant uses water to circulate the glucose around the plant to deliver the glucose to where it's needed.
What happened to water?

**LLM Sample Output:**
*Water is converted in story A.*

Figure 15: Sample prompt used for conversion story selection step of unstructured in-context learning (ICL-U) baseline on Tiered-ProPara. Some redundant sections of the prompt are omitted (indicated with "..." and in section headings).

## Tiered-ProPara - ICL-U Baseline - Conversion Sentence Selection

**Demonstration Prompts (1 of 4):**
Story A:
1. Wind and water erode rocks.
2. The wind and water pick up sediment that came off the rocks.
3. Sediment is carried to an area by the wind, water, air, or ice.
4. Sediment is deposited on the same spot many times.
5. The sediment piles on top of other sediment for a long time.
6. The sediment near the bottom is compressed by the weight of newer sediment.
7. The sediment becomes sedimentary rock as it is pushed together by the heavy weight.
Story B:
1. Waves hit the coast.
2. Waves have energy that pushes the water and sediment.
3. The sediment can help erode land like sandpaper.
4. Water washes the sediment back.
5. Some sediment is left as sand.
What happened to sediment?
Sediment is converted in sentence 7 in story A.

**Test Prompt:**
Story A:
1. Coal is heated in the boiler.
2. The water tank over the boiler is heated.
3. Creates steam.
4. The steam is funneled to the piston.
5. Piston uses the steam as energy.
6. The piston causes the crankshaft to move.
Story B:
1. Plants obtain water through the soil they are growing in.
2. The plants roots absorb the water from the soil.
3. Transport the water to the parts of the plant where the water is needed.
4. The plant uses the water as part of the photosynthesis process.
5. The plant creates food, called glucose, for itself.
6. The plant uses water to circulate the glucose around the plant to deliver the glucose to where it's needed.
What happened to water?

**LLM Sample Output:**
*Water is converted in sentence 3 in story A.*

Figure 16: Sample prompt used for conversion sentence selection step of unstructured in-context learning (ICL-U) baseline on Tiered-ProPara. Some redundant sections of the prompt are omitted (indicated with "..." and in section headings).

## Tiered-ProPara - ICL-U Baseline - Physical State Prediction

**Demonstration Prompts (1 of 4):**
Story A:
1. Wind and water erode rocks.
2. The wind and water pick up sediment that came off the rocks.
3. Sediment is carried to an area by the wind, water, air, or ice.
4. Sediment is deposited on the same spot many times.
5. The sediment piles on top of other sediment for a long time.
6. The sediment near the bottom is compressed by the weight of newer sediment.
7. The sediment becomes sedimentary rock as it is pushed together by the heavy weight.
Story B:
1. Waves hit the coast.
2. Waves have energy that pushes the water and sediment.
3. The sediment can help erode land like sandpaper.
4. Water washes the sediment back.
5. Some sediment is left as sand.
What happened to sediment?
Sediment is converted to sedimentary rock.

**Test Prompt:**
Story A:
1. Coal is heated in the boiler.
2. The water tank over the boiler is heated.
3. Creates steam.
4. The steam is funneled to the piston.
5. Piston uses the steam as energy.
6. The piston causes the crankshaft to move.
Story B:
1. Plants obtain water through the soil they are growing in.
2. The plants roots absorb the water from the soil.
3. Transport the water to the parts of the plant where the water is needed.
4. The plant uses the water as part of the photosynthesis process.
5. The plant creates food, called glucose, for itself.
6. The plant uses water to circulate the glucose around the plant to deliver the glucose to where it's needed.
What happened to water?

**LLM Sample Output:**
*Water is converted to steam.*

Figure 17: Sample prompt used for conversion entity prediction step of unstructured in-context learning (ICL-U) baseline on Tiered-ProPara. Some redundant sections of the prompt are omitted (indicated with "..." and in section headings).