# OpenReview forum: "From Heuristic to Analytic: Cognitively Motivated Strategies for Coherent Physical Commonsense Reasoning"
_EMNLP/2023/Conference — EMNLP 2023 Main_

### Official Review · Reviewer_5psW · 2023-08-02

**Soundness:** 4

**Excitement:**

3: Ambivalent: It has merits (e.g., it reports state-of-the-art results, the idea is nice), but there are key weaknesses (e.g., it describes incremental work), and it can significantly benefit from another round of revision. However, I won't object to accepting it if my co-reviewers champion it.

**Paper Topic And Main Contributions:**

The paper proposes heuristic-analytic reasoning (HAR) strategies to improve coherent physical commonsense reasoning in pre-trained language models (PLMs). It takes inspiration from dual process theories in cognitive psychology, where humans use quick heuristic thinking to make decisions, then slower analytic thinking to rationalize them. The paper implements HAR for fine-tuning and in-context learning with PLMs. In fine-tuning, models make high-level plausibility judgments on stories first, then these decisions are used to mask irrelevant information and focus attention when making lower-level judgments about conflicting sentences and physical states. In in-context learning, PLMs are prompted to generate reasoning chains from high-level story selection to low-level physical state prediction. Experiments on TRIP and a novel Tiered-ProPara dataset show HAR achieves promising performance compared to baselines. Analysis reveals that improved coherence is enabled by more faithful attention to relevant context when generating each step of reasoning under HAR. The key contribution of this paper is demonstrating that human-inspired heuristic and analytic reasoning strategies can mitigate incoherence and improve trustworthiness in PLMs for physical commonsense reasoning.

**Reasons To Accept:**

1. The authors demonstrate that PLMs' reasoning capabilities can benefit from incorporating complete cognitive processes containing both heuristic and analytic thinking. Experiments on fine-tuning and in-context learning with PLMs show that passing information from high-level to low-level tasks helps PLMs achieve better performance on low-level tasks.
2. The authors refashion the Propara dataset into a Tiered-Propara dataset and provide ample experimental results.
3. For in-context learning, the authors delve into the internals of LLMs. Utilizing the open-source LLaMA-65B, the authors visualize the Faithful Attention within the LLM, clearly and effectively displaying how HAR aids LLMs in focusing on the correct content.

**Reasons To Reject:**

1. The innovativeness of HAR is limited to some extent. On the one hand, HAR for PLM fine-tuning can be considered as constructing a pipeline of subtasks, so its performance gains are unsurprising. On the other hand, HAR for PLM in-context learning is a kind of "reverse" chain-of-thought method tailored for physical commonsense reasoning, obtaining the result first and then analyzing the process.
2. The idea behind HAR is similar to the paper "Language Modeling with Latent Situations" published in December 2022, both utilizing intermediate state inference to enhance reasoning capabilities, and applying both fine-tuning and prompting. The authors should include a comparison to that work and analyze the unique aspects of HAR in more depth.

**Reproducibility:**

4: Could mostly reproduce the results, but there may be some variation because of sample variance or minor variations in their interpretation of the protocol or method.

**Reviewer Confidence:**

4: Quite sure. I tried to check the important points carefully. It's unlikely, though conceivable, that I missed something that should affect my ratings.

---

> ### Author Rebuttal · Authors · 2023-08-28
>
> Thank you for the thoughtful comments and feedback! We appreciate your recognition of our thorough experiments and analysis, including our effort to reframe the ProPara dataset. We respond to your concerns below.
>
> **W1: Innovativeness of HAR**
>
> While these approaches may seem simple, they lead to novel and valuable insights. HAR enables performance improvements by refocusing LLMs to use the correct context to generate more coherent reasoning chains, a phenomenon which (to our knowledge) hasn’t yet been studied in in-context learning with LLMs. This finding aligns with dual process theories from cognitive psychology research (Wason, 1974; Evans 1976; Evans 1984), providing hope and motivation for future work to further explore cognitively-motivated strategies to enable more coherent reasoning in LLMs, and mitigate timely issues like hallucination which limit their practical usefulness. We will better emphasize this in our revision.
>
> **W2: Missing Paper “Language Modeling with Latent Situations”**
>
> While our paper works on some of the same problems as this paper which recently appeared in Findings of ACL 2023, our approach is quite different. Like this work, we propose strategies for applying LLMs to downstream reasoning tasks in both in-context learning and fine-tuning settings. Their work shows that learning (low-level) world state representations improves coherent reasoning in LLMs. Our work, somewhat complementary, shows that supporting the (analytic) prediction of these low-level physical states is improved by conditioning it with higher-level (heuristic) predictions about texts, and this is enabled by a shift in attentional patterns when solving tasks.
>
> We would have liked to include this paper in our results, but it does not report the same evaluation metrics on TRIP as we do. However, we will certainly add this paper to our Related Work section now that it is published. Thank you for pointing it out!

---

### Official Review · Reviewer_CnKW · 2023-08-04

**Soundness:** 3

**Excitement:**

4: Strong: This paper deepens the understanding of some phenomenon or lowers the barriers to an existing research direction.

**Paper Topic And Main Contributions:**

This paper introduces the concept of heuristic-analytic reasoning (HAR) in pre-trained language models (PLMs) for coherent commonsense reasoning. The authors propose two strategies to incorporate HAR into PLM fine-tuning and in-context learning. They evaluate the effectiveness of HAR on two benchmark tasks, Tiered Reasoning for Intuitive Physics (TRIP) and Tiered-ProPara, and compare it to baseline approaches. The results demonstrate that HAR leads to substantial enhancements in reasoning coherence, surpassing competitive baselines. Additionally, the approach enables PLMs to focus more faithfully on relevant language context during reasoning.

**Questions For The Authors:**

QA. Why is HAR more suitable to improve performance on physics tasks than other prompting methods (such as CoT)? Given its resemblance to CoT prompting, I would like to see some similar techniques (e.g., those mentioned in Section 2) as baselines in Section 5.

QB. In Table 3, the ICL-U method exhibits superior accuracy compared to ICL-HAR in both TRIP (70.4/55.6) and Tiered-ProPara (51.2/41.8). Can you provide an explanation for this discrepancy in the paper?

QC. How do you deal with the problem mentioned in the limitations that model outputs may be influenced by decisions/errors on higher-level tasks? Will it affect the effectiveness of the model?

**Reasons To Accept:**

1. Introducing heuristic-analytic reasoning to PLMs for coherent commonsense reasoning, inspired by human cognition, offers valuable insights for enhancing AI reasoning.
2. Demonstrating significant performance improvements with HAR on TRIP and Tiered-ProPara benchmarks highlights its effectiveness in enhancing reasoning coherence, for both fine-tuning and in context learning using HAR.
3. Coherent reasoning is an important research question of PLMs, this paper fills the void and uses consistency and verifiability to evaluate lower-level steps.

**Reasons To Reject:**

1. The HAR method in this article shares some similarities with step-by-step thinking (a chain-of-thought with fixed inference steps), but it significantly differs from the dual pathway in cognitive science, which involves combining intuition and rationality. As a result, the motivation for this article may not be appropriate.
2. As mentioned in the limitation section, this approach potentially suffers from cascading errors in PLMs' reasoning, propagating from higher-level to lower-level tasks. Now that there are works on reflective/ iterative reasoning, the novelty of this work seems a little limited.
3. I would like to see a more in-depth analysis of this task. This work mainly focuses on physical commonsense reasoning task. However, dual process theory is a common way of thinking for humans and is not limited to specific task scenarios. So it would be better to clearly point out and address the specific challenges of physical scenarios.

**Reproducibility:**

4: Could mostly reproduce the results, but there may be some variation because of sample variance or minor variations in their interpretation of the protocol or method.

**Reviewer Confidence:**

4: Quite sure. I tried to check the important points carefully. It's unlikely, though conceivable, that I missed something that should affect my ratings.

**Typos Grammar Style And Presentation Improvements:**

- In Line 374, the authors mentioned that former PLM is limited to a context length of 2048 tokens, which is not true for InstructGPT (4096 tokens).

---

> ### Author Rebuttal · Authors · 2023-08-28
>
> Thanks for your thoughtful comments and feedback! We appreciate your recognition of the valuable insights our work can bring for AI-based reasoning, and respond to your concerns below.
>
> **W1: Cognitive Motivations of HAR**
>
> Sorry, perhaps there is a misunderstanding here. HAR is motivated by the dual process theory which utilizes intuition to remove irrelevant information and process low-level rationalizations. In the paper “Heuristic and analytic processes in reasoning” (Evans, 1984), Evans highlighted “a distinction between heuristic processes which select items of task information as ‘relevant’, and analytic processes which operate on the selected items to generate inferences or judgements.” Works cited in Section 1 review a breadth of cognitive psychology research that supports this theory.
>
> Similar to Evans’ description, HAR first makes an intuitive prediction on a high-level task which acts as a heuristic for rationalizing a low-level task. Taking the TRIP dataset as an example, HAR judges whether a story is plausible or not in the first step, identifies the conflicting sentences in the second step, and then rationalizes the physical states that cause the conflict in the third step. Our attention analysis experiments in Section 5.2 verify that this occurs due to the model’s stronger focus on relevant information during reasoning, which resembles the above synergy of heuristic and analytic processes in human cognitive psychology. Therefore, we think our motivation is appropriate and verified.
>
> Furthermore, it is worth noting that HAR differs from traditional chain-of-thought in prompting LLMs (CoT). We discuss this in more detail in our response to your QA.
>
> **W2: Cascading Errors**
>
> Yes, as discussed in our Limitations section, the cascading errors in PLMs’ reasoning is a possible limitation. It is indeed possible that various non-greedy generation or reasoning strategies may be applied on top of HAR to mitigate this limitation, such as beam search or [tree-of-thought](https://arxiv.org/abs/2305.10601) to generate multiple candidate rationalizations and choose one based on LLM likelihoods. Such methods to reduce cascading error are orthogonal to our efforts, as they could theoretically be applied to any of the comparison approaches used in this work (not just HAR). As our goal was to demonstrate that heuristic decisions can be used to improve performance on analytic rationalizations in PLMs, we chose not to explore these approaches here as they would increase the moving parts in our study, and possibly distract from this key message.
>
> Nonetheless, we show that even with possible cascading error, the model still outperforms all baselines by a great margin. The gradient-based fine-tuning model achieved state-of-the-art results on TRIP despite this effect. We’ll discuss this issue further in our revised version.
>
> And while these approaches may seem simple, they lead to novel and valuable insights. HAR enables performance improvements by refocusing LLMs to use the correct context to generate more coherent reasoning chains, a phenomenon which (to our knowledge) hasn’t yet been studied in in-context learning with LLMs. This finding aligns with dual process theories from cognitive psychology research (Wason, 1974; Evans 1976; Evans 1984), providing hope and motivation for future work to further explore cognitively-motivated strategies to enable more coherent reasoning in LLMs, and mitigate timely issues like hallucination which limit their practical usefulness.
>
> **W3: Analysis of Tasks**
>
> Good point! We agree that dual process theory is a common way of thinking for humans and is not limited to any specific task. We choose physical commonsense reasoning as an example because it involves high-level conclusions (e.g. plausibility of a story or other decisions about texts describing actions) and low-level predictions (e.g. physical states of objects) and there are datasets that contain annotations for different levels. Additionally, our paper addresses one of the specific challenges in physical scenarios (improve coherence and low-level physical state predictions), which has a possibly broad impact in embodied AI, where human users will communicate with agents about actions, and expect that agents have a coherent understanding of them in terms of how they will impact the surrounding environment.
>
> To the best of our knowledge, there are few available datasets for other scenarios that can be used in such a tiered reasoning setup (we have searched extensively). We acknowledge the importance of creating such datasets and encourage future works to further explore in this direction to better characterize the space of tasks where heuristic-analytic reasoning can be beneficial. We will add more discussion about this in our revision.
>
> **QA: Comparison to Chain-of-Thought**
>
> We provide some explanation above as to why we focused on physical commonsense. As for the resemblance to CoT prompting, there’s an important distinction between our method (HAR) and traditional Chain-of-Thought (CoT). Traditional CoT breaks down a hard problem into smaller subproblems to improve complex reasoning, such as for multi-step math problems. In physical commonsense, low-level atomic physical states are the most difficult for AI models to predict, and they are difficult to further decompose. As such, **we argue that for these kind of problems, the key for solving them is not decomposing them, but rather helping the model make predictions over the correct context (e.g., pay more attention to the correct context and remove irrelevant information).** HAR achieves this by conditioning rationalization predictions with information from higher-level heuristic predictions. We will clarify this in the revised version.
>
> Nonetheless, we attempted to integrate CoT into our framework by augmenting the ICL-U prompts for each of the 3 sub-tasks (i.e., story selection, sentence selection, and state prediction) with free-text explanations. These explanations are generated by the LLM by prompting it with “Think step by step” after the sub-task context for each in-context demonstration and the testing example being solved, then they’re appended to the prompt. This gave us a few-shot CoT method that applies to our task. The results below show that applying traditional CoT (ICL-CoT) doesn’t improve performance much. Further, we observed that the generated reasoning explanations tend to iterate through sentences in stories to do step by step reasoning upon them, which doesn’t add useful information to the problem or simplify it- the model still needs to make predictions on lower-level tasks from a large context with a lot of irrelevant information. Therefore, we proposed HAR, a method that can use high-level (heuristic) predictions to focus on the correct context for low-level (analytic) predictions.
>
> *TRIP (InstructGPT)*
> | Method | Accuracy | Consistency | Verifiability |
> |--|--|--|--|
> | ICL-U | 70.9 | 40.7 | 7.1 |
> | ICL-CoT | 70.7 | 39.6 | 8.3 |
> | ICL-HAR | **72.6** | **47.9** | **23.9** |
>
> *TRIP (LLaMA)*
> | Method | Accuracy | Consistency | Verifiability |
> |--|--|--|--|
> | ICL-U | **70.4** | 42.3 | 14.8 |
> | ICL-CoT | 62.0 | 43.7 | 14.1 |
> | ICL-HAR | 55.6 | **44.4** | **35.2** |
>
> *Tiered-Propara (InstructGPT)*
> | Method | Accuracy | Consistency | Verifiability |
> |--|--|--|--|
> | ICL-U | **54.9** | 17.4 | 5.2 |
> | ICL-CoT | 52.1 | 23.9 | 12.2 |
> | ICL-HAR | **54.9** | **31.5** | **20.7** |
>
> *Tiered-Propara (LLaMA)*
> | Method | Accuracy | Consistency | Verifiability |
> |--|--|--|--|
> | ICL-U | **51.2** | 3.8 | 1.4 |
> | ICL-CoT | 48.8 | 4.7 | 1.9 |
> | ICL-HAR | 41.8 | **17.8** | **13.1** |
>
> We will add this result into the revised version.
>
> **QB: Accuracy Drop in ICL-HAR with LLaMA**
>
> Good observation! Although ICL-U is more accurate than ICL-HAR in some cases, we would highlight that the focus of these tasks and our work is to improve the *coherence* of reasoning, which accuracy does not capture (only measures performance on the end task of story selection). Instead, our goal is to optimize **consistency** and **verifiability**, each of which already require end-task predictions to be accurate as a prerequisite, and measure the correctness of the sentences and physical states identified by the model to rationalize them. In fact, it is not ideal for accuracy to far exceed these metrics, as this indicates the model is making a correct prediction on the end task without valid underlying support (incoherent reasoning). Therefore, a drop in accuracy is not problematic in these tasks. We will better clarify the significance of each evaluation metric in our revision.
>
> That said, this drop only happens with LLaMA-65B, and we think this is because it’s much smaller than InstructGPT (175B), so it is more sensitive to long prompts and generations. A quick fix could be to sequentially prompt the LLM multiple times with shorter prompts for each sub-task. We presented results for an approach like this in Appendix A.2, and they don’t exhibit such a drop in accuracy.
>
> **QC: Cascading Errors**
>
> Yes, as discussed in our Limitations section, the cascading errors in PLMs’ reasoning is a possible limitation. It is indeed possible that various non-greedy generation or reasoning strategies may be applied on top of HAR to mitigate this limitation, such as beam search or [tree-of-thought](https://arxiv.org/abs/2305.10601) to generate multiple candidate rationalizations and choose one based on LLM likelihoods. Such methods to reduce cascading error are orthogonal to our efforts, as they could theoretically be applied to any of the comparison approaches used in this work (not just HAR). As our goal was to demonstrate that heuristic decisions can be used to improve performance on analytic rationalizations in PLMs, we chose not to explore these approaches here as they would increase the moving parts in our study, and possibly distract from this key message.
>
> Nonetheless, we show that even with possible cascading error, the model still outperforms all baselines by a great margin. The gradient-based fine-tuning model achieved state-of-the-art results on TRIP despite this effect. We’ll discuss this issue further in our revised version.
>
> **Typos: Model Context Length**
>
> Thank you for pointing out this typo! We should use “latter” instead of “former” to refer to LLaMA-65B model. InstructGPT is indeed having a context length of 4096 tokens, but LLaMA-65B only has a context length of 2048 tokens so we created a filtered version of TRIP to make the prompt shorter for the familiarization. We will fix this in our revision.

---

### Official Review · Reviewer_nRZC · 2023-08-11

**Soundness:** 3

**Excitement:**

4: Strong: This paper deepens the understanding of some phenomenon or lowers the barriers to an existing research direction.

**Paper Topic And Main Contributions:**

This paper proposes one fine-tuning and one in-context learning frameworks to improve the reasoning consistency and verifiability on physical commonsense reasoning benchmarks (TRIP and ProPara). The heuristic-analytical reasoning (HAR) framework is inspired by human cognitive processes of reasoning, where a heuristic greedy decision is firstly arrived based on simple associative features before engaging in an analytical and rational process. The framework is implemented by breaking the reasoning task into multiple sub-steps (e.g. identify story plausibility -> detect conflicting sentence -> focus on physical state). The HAR significantly improves the coherence of model decisions, and that the improvement is attributed to a faithful attention to relevant language context.

**Questions For The Authors:**

A. I am unclear about line 390 to line 393 even after looking at the appendix. How are the 3 extracted predictions on each testing exampled combined? How is the metrics calculated?

**Reasons To Accept:**

1. The principled framework significantly improves the reasoning consistency on physical commonsense reasoning benchmarks.

2. The attention weights analysis effectively explains why the framework helps: the guided procedure successfully shifts model attention to the key aspects of the long discourse.

**Reasons To Reject:**

1. Generalizability: both fine-tuning and in-context learning strategies seem to be tailored for shifting model attention to a smaller chunk of key information, which is confirmed by the attention weight analysis. This makes me to worry to what extent this method could be generalized to other datasets where the information is not presented in contrastive pairs, and where the conflicting information is not restricted to one or two sentences but widely spread in the entire passage. For example, the task of identifying conflicting sentences might have over-simplified the reasoning task by taking the short-cut to ignore lots information, which just happens to be trivial in these specific datasets.

2. The developed strategies, while interesting, might just be marginally relevant to the cognitive process of heuristic / analytical dual passes of human reasoning. The heuristic reasoning process is more related to the information being utilized and the amount of attention paid to more fine-grained details. For instance, in online language comprehension, comprehenders might ignore fine-grained syntactic structured and rely on the semantic meaning of the words and their prior knowledge to interpret "the hearty meal was devouring..." as "the hearty meal was devoured..." (Kim and Osterhout, 2005). However, the heuristic process is less concerned with gratuity of final decision, as implied by the HAR model. The HAR framework breaks the reasoning tasks into multiple sub-tasks, where the gratuity of the decision gradually becomes finer-grained. This might be better characteristics as step-by-step chain of reasoning rather than heuristic decision-making.

3. The ICL-HAR, while improving consistency and verifiability, has greatly impedes the accuracy scores (dropping from 70.4 to 55.6 on TRIP).  This should be discussed or at least acknowledged in the main text in more detail.


**Reproducibility:**

4: Could mostly reproduce the results, but there may be some variation because of sample variance or minor variations in their interpretation of the protocol or method.

**Reviewer Confidence:**

3: Pretty sure, but there's a chance I missed something. Although I have a good feel for this area in general, I did not carefully check the paper's details, e.g., the math, experimental design, or novelty.

---

> ### Author Rebuttal · Authors · 2023-08-28
>
> Thank you for the thoughtful comments and suggestions! We appreciate your recognition of our models’ performance and the effectiveness of our analysis, and respond to your review below.
>
> **W1: Generalizability**
>
> This is a good point! While the datasets we studied highlight the benefit of heuristic-analytic reasoning due to critical information being localized in small parts of text, this is not a requirement for HAR to be applied, especially for the in-context learning implementation which is soft and can still take the entire context into account as more heuristic information is introduced. Although not the focus of our work, it would be interesting for future work to study attention patterns in these sorts of tasks where critical information can be more widespread.
>
> Furthermore, the type of heuristic-analytic reasoning we study in this work can actually be quite abundant in our real world. For example, when we are trying to identify a bug in our code, we often utilize a heuristic intuition on where the bug may be located based on sparse information. Then, we will use analytic reasoning to find out the bug (e.g., trace code with debugger, add print statements, etc.) and fix it.
>
> We will discuss this issue of generalizability in more detail in our revision.
>
> **W2: Relevance to Cognitive Processes**
>
> This work specifically focuses on how chaining together these dual processes may support coherent reasoning in LLMs. HAR is motivated by the dual process theory which utilizes intuition to remove irrelevant information and process low-level rationalizations. In the paper “Heuristic and analytic processes in reasoning” (Evans, 1984), Evans highlighted “a distinction between heuristic processes which select items of task information as ‘relevant’, and analytic processes which operate on the selected items to generate inferences or judgements.” Works cited in Section 1 review a breadth of cognitive psychology research that supports this theory.
>
> Similar to Evans’ description, HAR first makes an intuitive prediction on a high-level task which acts as a heuristic for rationalizing a low-level task. Taking the TRIP dataset as an example, HAR judges whether a story is plausible or not in the first step, identifies the conflicting sentences in the second step, and then rationalizes the physical states that cause the conflict in the third step. Our attention analysis experiments in Section 5.2 verify that this occurs due to the model’s stronger focus on relevant information during reasoning, which resembles the above synergy of heuristic and analytic processes in human cognitive psychology.
>
> Furthermore, it may be worth noting that our method (HAR) differs from traditional Chain-of-Thought (CoT) used for step-by-step reasoning. Traditional CoT breaks down a hard problem into smaller subproblems to improve complex reasoning, such as for multi-step math problems. In physical commonsense, low-level atomic physical states are the most difficult for AI models to predict, and they are difficult to further decompose.
>
> Nonetheless, we did attempt to integrate CoT into our framework by augmenting the ICL-U prompts for each of the 3 sub-tasks (i.e., story selection, sentence selection, and state prediction) with free-text explanations. These explanations are generated by the LLM by prompting it with “Think step by step” after the sub-task context for each in-context demonstration and the testing example being solved, then they’re appended to the prompt. This gave us a few-shot CoT method that applies to our task. The results below show that applying traditional CoT (ICL-CoT) doesn’t improve performance much. Further, we observed that the generated reasoning explanations tend to iterate through sentences in stories to do step by step reasoning upon them, which doesn’t add useful information to the problem or simplify it- the model still needs to make predictions on lower-level tasks from a large context with a lot of irrelevant information. Therefore, we proposed HAR, a method that can use high-level (heuristic) predictions to focus on the correct context for low-level (analytic) predictions.
>
> *TRIP (InstructGPT)*
> | Method | Accuracy | Consistency | Verifiability |
> |--|--|--|--|
> | ICL-U | 70.9 | 40.7 | 7.1 |
> | ICL-CoT | 70.7 | 39.6 | 8.3 |
> | ICL-HAR | **72.6** | **47.9** | **23.9** |
>
> *TRIP (LLaMA)*
> | Method | Accuracy | Consistency | Verifiability |
> |--|--|--|--|
> | ICL-U | **70.4** | 42.3 | 14.8 |
> | ICL-CoT | 62.0 | 43.7 | 14.1 |
> | ICL-HAR | 55.6 | **44.4** | **35.2** |
>
> *Tiered-Propara (InstructGPT)*
> | Method | Accuracy | Consistency | Verifiability |
> |--|--|--|--|
> | ICL-U | **54.9** | 17.4 | 5.2 |
> | ICL-CoT | 52.1 | 23.9 | 12.2 |
> | ICL-HAR | **54.9** | **31.5** | **20.7** |
>
> *Tiered-Propara (LLaMA)*
> | Method | Accuracy | Consistency | Verifiability |
> |--|--|--|--|
> | ICL-U | **51.2** | 3.8 | 1.4 |
> | ICL-CoT | 48.8 | 4.7 | 1.9 |
> | ICL-HAR | 41.8 | **17.8** | **13.1** |
>
> We will add this result into the revised version.
>
> **W3: Accuracy Drop in ICL-HAR with LLaMA**
>
> Good observation! Although ICL-U is more accurate than ICL-HAR in some cases, we would highlight that the focus of these tasks and our work is to improve the *coherence* of reasoning, which accuracy does not capture (only measures performance on the end task of story selection). Instead, our goal is to optimize **consistency** and **verifiability**, each of which already require end-task predictions to be accurate as a prerequisite, and measure the correctness of the sentences and physical states identified by the model to rationalize them. In fact, it is not ideal for accuracy to far exceed these metrics, as this indicates the model is making a correct prediction on the end task without valid underlying support (incoherent reasoning). Therefore, a drop in accuracy is not problematic in these tasks. We will better clarify the significance of each evaluation metric in our revision.
>
> That said, this drop only happens with LLaMA-65B, and we think this is because it’s much smaller than InstructGPT (175B), so it is more sensitive to long prompts and generations. A quick fix could be to sequentially prompt the LLM multiple times with shorter prompts for each sub-task. We presented results for an approach like this in Appendix A.2, and they don’t exhibit such a drop in accuracy.
>
> **QA: Metric Calculation**
>
> Sorry for glossing over the details here. As shown in Figures 12-14 and 15-17, we prompt LLMs separately for 3 tiered tasks of story, sentence, and low-level physical state classification. Taking TRIP as an example (Figures 12-14), we use regular expressions to extract the LLM’s predicted plausible story, conflicting sentences, and underlying physical state labels automatically from generated text. These 3 predictions make up a complete reasoning chain on the TRIP problem, as introduced in [Storks et al. (2021)](https://aclanthology.org/2021.findings-emnlp.422.pdf). Once we have extracted such reasoning chains from each example in the evaluation dataset, we calculate the metrics. Accuracy is taken as the proportion of chains such that the chosen plausible story is correct. Consistency is taken as the proportion of chains such that the predicted pair of conflicting sentences is additionally correct. Lastly, verifiability is taken as the proportion of chains such that the underlying physical states within those correct conflicting sentences are also correct (at least one non-default state is predicted in each sentence, and all such predicted states are correct). As such, it is always the case that accuracy >= consistency >= verifiability, and each of these metrics evaluates lower-level pieces of evidence in the reasoning chain to target the coherence of reasoning.
>
> We will clarify this in our revision.

---

### Official Review · Reviewer_KHzS · 2023-08-11

**Typos Grammar Style And Presentation Improvements:** 1. From my perspective, I think that …
**Soundness:** 4

**Excitement:**

3: Ambivalent: It has merits (e.g., it reports state-of-the-art results, the idea is nice), but there are key weaknesses (e.g., it describes incremental work), and it can significantly benefit from another round of revision. However, I won't object to accepting it if my co-reviewers champion it.

**Paper Topic And Main Contributions:**

The paper explores strategies to improve the coherence and trustworthiness of reasoning by large pre-trained language models (PLMs) on physical commonsense reasoning tasks. It takes inspiration from dual-process theories in cognitive psychology, which posit humans use fast, intuitive heuristic thinking and slower, logical analytic thinking. Experiments on two physical commonsense benchmarks, TRIP, and Tiered-ProPara, show HAR improves coherence metrics like consistency and verifiability.

**Questions For The Authors:**

1. #305 - #307, authors include previously published RoBERTa results from Storks et al. (2021), CGLI (Ma et al., 2022), and BT (Richardson et al., 2022, are these results reported under the same settings?

**Reasons To Accept:**

1. Novelty: Drawing the idea from cognitive psychology, this paper explores strategies to improve the coherence and trustworthiness of reasoning. The key ideas are using human-inspired heuristics and then analytic reasoning steps to improve the coherence of PLMs, testing this on physical commonsense tasks, and showing it strengthens rationalization by attention.
2. The authors innovatively propose Tiered-ProPara tasks for coherent physical commonsense reasoning, which leaves room for further exploration.
3. The paper is generally well-written, and most parts of the paper are easy to follow.

**Reasons To Reject:**

1. The proposed heuristic-analytic reasoning strategies rely on making sequential predictions, which could cause cascading errors if the model makes a mistake on an early heuristic prediction.
2. Some of the gains may come from simply breaking down a complex task rather than the proposed psychology-inspired heuristic-analytic duality. Testing on broader tasks could help isolate the benefits.
3. Although the authors claim that "verifiability from 7.1% up to 23.9%, over a 200% improvement" for InstructGPT and "5.2% to 20.7%, nearly a 300% improvement" for LLaMA, it seems that it will hamper the accuracy scores, and need further discussion. Also, the sample size seems to be not big enough, and more extensive evaluations could help validate the results and trends.

**Reproducibility:**

4: Could mostly reproduce the results, but there may be some variation because of sample variance or minor variations in their interpretation of the protocol or method.

**Reviewer Confidence:**

3: Pretty sure, but there's a chance I missed something. Although I have a good feel for this area in general, I did not carefully check the paper's details, e.g., the math, experimental design, or novelty.

---

> ### Author Rebuttal · Authors · 2023-08-28
>
> Thank you for your thoughtful and constructive feedback, and for recognizing the novelty of our work in using insight from cognitive psychology to improve coherence and trustworthiness of reasoning in LLMs, and in proposing a reframed ProPara dataset. We respond to concerns about the work below.
>
> **W1: Regarding Cascading Errors**
>
> Yes, as discussed in our Limitations section, the cascading errors in PLMs’ reasoning is a possible limitation. It is indeed possible that various non-greedy generation or reasoning strategies may be applied on top of HAR to mitigate this limitation, such as beam search or [tree-of-thought](https://arxiv.org/abs/2305.10601) to generate multiple candidate rationalizations and choose one based on LLM likelihoods. Such methods to reduce cascading error are orthogonal to our efforts, as they could theoretically be applied to any of the comparison approaches used in this work (not just HAR). As our goal was to demonstrate that heuristic decisions can be used to improve performance on analytic rationalizations in PLMs, we chose not to explore these approaches here as they would increase the moving parts in our study, and possibly distract from this key message.
>
> Nonetheless, we show that even with possible cascading error, the model still outperforms all baselines by a great margin. The gradient-based fine-tuning model achieved state-of-the-art results on TRIP despite this effect. We’ll discuss this issue further in our revised version.
>
> **W2: Gains from Breaking Down the Complex Task**
>
> This is an important point which was also at the front of our minds when designing experiments! This possibility motivated our unstructured baselines which still broke the complex tasks down into smaller parts (e.g., by prompting LLMs with smaller separate prompts for each sub-task), but did not condition rationalization steps with higher-level heuristic decisions as we did in HAR methods. By using this as a comparison, we demonstrated that a large amount of performance gains indeed come from connecting heuristic and analytic reasoning steps as inspired by dual process theories in cognitive psychology. Furthermore, we empirically verify that this happens due to refocusing of model attention on the most relevant information during reasoning, which resembles what is theorized for human reasoning.
>
> Testing on broader tasks would have been ideal, but there are not many available resources for this type of evaluation of coherent reasoning. While this approach can theoretically be applied to any language problems with higher-level decisions supported by more complex rationalizations, there are few datasets for this (partly why we created Tiered-ProPara in this work). Despite this, it is worth noting that these efforts in physical commonsense can have a broad impact, as coherent reasoning will become especially important with the development of embodied AI, as a human user will need to trust that an embodied agent deeply understands the physical actions they communicate with the agent about, and how they will impact the world around them.
>
> We also want to add here that traditional chain-of-thought (CoT) has this motivation of breaking down complex tasks to improve performance, but method (HAR) differs from traditional Chain-of-Thought (CoT). Traditional CoT breaks down a hard problem into smaller subproblems to improve complex reasoning, such as for multi-step math problems. In physical commonsense, low-level atomic physical states are the most difficult for AI models to predict, and they are difficult to further decompose.
>
> Nonetheless, we attempted to integrate CoT into our framework by augmenting the ICL-U prompts for each of the 3 sub-tasks (i.e., story selection, sentence selection, and state prediction) with free-text explanations. These explanations are generated by the LLM by prompting it with “Think step by step” after the sub-task context for each in-context demonstration and the testing example being solved, then they’re appended to the prompt. This gave us a few-shot CoT method that applies to our task. The results below show that applying traditional CoT (ICL-CoT) doesn’t improve performance much. Further, we observed that the generated reasoning explanations tend to iterate through sentences in stories to do step by step reasoning upon them, which doesn’t add useful information to the problem or simplify it- the model still needs to make predictions on lower-level tasks from a large context with a lot of irrelevant information. Therefore, we proposed HAR, a method that can use high-level (heuristic) predictions to focus on the correct context for low-level (analytic) predictions.
>
> *TRIP (InstructGPT)*
> | Method | Accuracy | Consistency | Verifiability |
> |--|--|--|--|
> | ICL-U | 70.9 | 40.7 | 7.1 |
> | ICL-CoT | 70.7 | 39.6 | 8.3 |
> | ICL-HAR | **72.6** | **47.9** | **23.9** |
>
> *TRIP (LLaMA)*
> | Method | Accuracy | Consistency | Verifiability |
> |--|--|--|--|
> | ICL-U | **70.4** | 42.3 | 14.8 |
> | ICL-CoT | 62.0 | 43.7 | 14.1 |
> | ICL-HAR | 55.6 | **44.4** | **35.2** |
>
> *Tiered-Propara (InstructGPT)*
> | Method | Accuracy | Consistency | Verifiability |
> |--|--|--|--|
> | ICL-U | **54.9** | 17.4 | 5.2 |
> | ICL-CoT | 52.1 | 23.9 | 12.2 |
> | ICL-HAR | **54.9** | **31.5** | **20.7** |
>
> *Tiered-Propara (LLaMA)*
> | Method | Accuracy | Consistency | Verifiability |
> |--|--|--|--|
> | ICL-U | **51.2** | 3.8 | 1.4 |
> | ICL-CoT | 48.8 | 4.7 | 1.9 |
> | ICL-HAR | 41.8 | **17.8** | **13.1** |
>
> We will add this result into the revised version.
>
> **W3: Accuracy Drop in ICL-HAR with LLaMA**
>
> Good observation! Although ICL-U is more accurate than ICL-HAR in some cases, we would highlight that the focus of these tasks and our work is to improve the *coherence* of reasoning, which accuracy does not capture (only measures performance on the end task of story selection). Instead, our goal is to optimize **consistency** and **verifiability**, each of which already require end-task predictions to be accurate as a prerequisite, and measure the correctness of the sentences and physical states identified by the model to rationalize them. In fact, it is not ideal for accuracy to far exceed these metrics, as this indicates the model is making a correct prediction on the end task without valid underlying support (incoherent reasoning). Therefore, a drop in accuracy is not problematic in these tasks. We will better clarify the significance of each evaluation metric in our revision.
>
> That said, this drop only happens with LLaMA-65B, and we think this is because it’s much smaller than InstructGPT (175B), so it is more sensitive to long prompts and generations. A quick fix could be to sequentially prompt the LLM multiple times with shorter prompts for each sub-task. We presented results for an approach like this in Appendix A.2, and they don’t exhibit such a drop in accuracy.
>
> Regarding the sample size possibly not being big enough, we do agree that TRIP and Tiered Propara datasets are small datasets (densely annotated data for physical commonsense is scarce and expensive to collect). However, both datasets have been used extensively in empirical research in prior work, and the test sets are sufficiently large for statistically significant results (TRIP has 351 examples and Tiered-ProPara has 214 examples after our augmentation). While the differences between most results are already quite large, we additionally ran a McNemar test and found statistically significant (p < 0.05) differences in consistency and verifiability for our ICL-HAR and ICL-U results on TRIP. As *p*-values were many orders of magnitude less than 0.05, we expect other results to show similar significance, and will complete the remaining analysis for our revised paper.
>
> **Q1: Settings for Past Work Comparisons**
>
> We made substantial efforts to ensure that all results were comparable to those from past work, including verifying the same testing examples were used and using a slight modification of CGLI as a baseline (FCGLI) and largely mirroring the training processes of these past works (Storks et al., 2021; Ma et al., 2022), including using the same RoBERTa backbone and total number of training epochs. For BT (Richardson et al., 2022), we reported the results from their paper, but there are some differences: they use T5 as the PLM, and some hyperparameters may be in slightly different ranges (e.g., number of training epochs). Several published papers from this line of work on physical commonsense also report these numbers from prior work.
>
> **Presentation Improvements: Abbreviations**
>
> Thank you for letting us know that the abbreviations caused some difficulty in understanding. We will remove unnecessary abbreviations in the revised version of the paper.

---

### Meta-Review · Area_Chair_S48R · 2023-09-19

**Recommendation:** 4

**Metareview:**

This paper takes inspiration from dual process theories in cognitive psychology, where humans use quick heuristic thinking to make decisions, then slower analytic thinking to rationalize them. While they use a simple approach (like a reverse CoT), it leads to valuable insights on their new benchmark "Tiered Propara". To understand these empirical gains, they visualize the Faithful Attention within the LLM, clearly and effectively displaying how their approach aids LLMs in focusing on the correct content.

Limitations:
While the simple approach presented in this paper works for the two benchmarks, fine-tuning and in-context learning strategies seem to be tailored for datasets where key information is localized, allowing for shifting model attention to a smaller chunk of key information. Thus, it remains to be empirically verified if this approach generalizes to other datasets where the key information is not localized. Further, errors in can be cascaded in this reverse CoT reasoning and this issue must be discussed in more detail.

Overall, the strengths outweigh the limitations. It would be valuable to include the reviewer's suggestions and rebuttal content in the camera ready.

---

### Decision · Program_Chairs · 2023-10-07

**Decision:**

Accept-Main

**Comment:**

This paper takes inspiration from dual process theories in cognitive psychology, where humans use quick heuristic thinking to make decisions, then slower analytic thinking to rationalize them. While they use a simple approach (like a reverse CoT), it leads to valuable insights on their new benchmark "Tiered Propara". To understand these empirical gains, they visualize the Faithful Attention within the LLM, clearly and effectively displaying how their approach aids LLMs in focusing on the correct content.

Limitations:
While the simple approach presented in this paper works for the two benchmarks, fine-tuning and in-context learning strategies seem to be tailored for datasets where key information is localized, allowing for shifting model attention to a smaller chunk of key information. Thus, it remains to be empirically verified if this approach generalizes to other datasets where the key information is not localized. Further, errors in can be cascaded in this reverse CoT reasoning and this issue must be discussed in more detail.

Overall, the strengths outweigh the limitations. It would be valuable to include the reviewer's suggestions and rebuttal content in the camera ready.